

# Consolidating the Randolph Glacier Inventory and the Glacier Inventory of China over the Qinghai-Tibetan Plateau and Investigating Glacier Changes Since the mid-20th Century

Xiaowan Liu[1,2,3], Zongxue Xu[1,2], Hong Yang[3,4], Xiuping Li[5], Dingzhi Peng[1,2]

[1]College of Water Sciences, Beijing Normal University, Beijing, 100875, China
[2]Beijiing Key Laboratory of Urban Hydrological Cycle and Sponge City Technology, Beijing, 100875, China
[3]Eawag, Swiss Federal Institute of Aquatic Science and Technology, 8600 Dübendorf, Switzerland
[4]Department of Environmental Science, MGU University of Basel, Petersplatz 1, 4001, Switzerland
[5]Institute of Tibetan Plateau Research, Chinese Academy of Sciences, Beijing, 100101, China

*Correspondence to*: Zongxue Xu (zongxuexu@vip.sina.com)

**Abstract.** Glacier retreat in the Qinghai-Tibetan Plateau (QTP), the 'third pole of the world', has attracted the attention of researchers worldwide. Glacier inventories in the 1970s and the 2000s provide valuable information to infer changes in individual glaciers. However, individual glacier volumes are either missing, incomplete or have large errors in these inventories, and thus, the use of these datasets to investigate changes in glaciers in QTP in the past few decades has become a challenge, particularly in the context of climate change. In this study, individual glacier volume data in the Randolph Glacier Inventory version 4.0 (RGI 4.0, 1970s) and the second Glacier Inventory of China (GIC-II, 2000s) are recalculated and consolidated using a slope-dependent algorithm based on elevation datasets for the QTP. The two consolidated inventories (The data are available under https://doi.org/10.11888/Glacio.tpdc.270390 (Liu, 2020). For the time of review, the data will be accessible through the following review link https://data.tpdc.ac.cn/en/data/4b88e394-0eb4-44c4-aa38-32aeb614daff/.) are validated by comparing the observed and estimated glacier data reported in the literature. The two consolidated glacier inventories are then compared for different mountains over the QTP to detect changes in glacier areas, volumes, fragmentation status, etc. during the past 3-4 decades. Based on the results, the slope-dependent algorithm performed well in computing individual glacier volumes and other elements, compared with the widely used volume-area scaling which often leads to overestimation in the interior Plateau and underestimation in other areas of the QTP in both RGI 4.0 and GIC-II. The comparison of the two inventories reveals a total area of glaciers in the QTP of approximately 59026.5 km$^2$ in the RGI 4.0 and 44301.2 km$^2$ in the GIC-II. The total glacier volume is 4045.9 km$^3$ in the GIC-II compared with 4716.7 km$^3$ in the RGI 4.0. The results suggest a significant retreat and melting of glaciers in the QTP. However, variations are observed in different glaciers. The Karakoram Mountains contain the largest number of surged glaciers, while the highest level of retreat is observed in the Gandise Mountains. An increase in the fragmentation index is observed in the northern mountains, particularly the Pamir Plateau, which displays the highest trends of glacier movement and deformation. The glacier volumes decrease mainly on south-westward aspects and increase to various extents on the other aspects of most mountains. The consolidation of the glacier inventories and the findings of the analysis performed in this study provide important databases for future glacier-related studies, particularly for investigating the effects of climate change on glaciers in the past and projecting future effects.

**Key words.** ice volume; RGI 4.0; GIC-II; glacier retreat; Himalayan Mountains; Qinghai-Tibetan Plateau



## 40 1 Introduction

### 41 1.1 Background

Glacier melting and retreat in the context of climate change have attracted increasing attention in the recent
years. Changes in glacier volumes and areas have been the focus of many studies due to their significant effects
on the hydrological cycle and feedback effects on climate circulation (Bolch, 2007; Zhu et al., 2018). The
Qinghai-Tibetan Plateau (QTP) has the largest ice storage, with an ice volume only inferior to polar regions (Liu
et al., 2000). Glaciers over the QTP are shrinking, particularly in recent decades, due to global warming (Kang et
al., 2010). According to the study by Qiu (2008), over eighty percent of glaciers in the QTP has been retreating
since the 1960s. A study led by a distinguished expert in glacier studies in China predicted that approximately
two-thirds of the glaciers in the QTP would disappear in 2050s with the current retreating rate (Yao et al.,
2012b). Glacier melting exerts substantial effects on river runoff. The most direct results include short-term
flooding, long-term drought, and intensifying/aggravating glacier-dammed lake, and disrupting the ecological
balance, among others (Benn et al., 2012; Kaushik et al., 2019; Li, 2012; Zhang et al., 2015). Moreover, as a link
to the global water cycle and energy transport, glacier retreat in the QTP may also alter the global climate. Thus,
an understanding of the changes in glaciers in the QTP is essential for both runoff and global change projections.

### 55 1.2 Literature review

Numerous efforts have been focused on glacier-related studies over the QTP in the past few decades. These studies
cover a wide range of aspects, such as the development of glacier monitoring and mapping technology, glacier
melt modelling, glacier mass balance calculations, the contribution of glacial meltwater to runoff, and the
interrelations between glacier retreat and climate change (Che et al., 2018; Gao et al., 2018; Zhu et al., 2018).
Many studies have focused on the ice mass balance in the future, causes of glacier retreat and effective adaptations.
However, due to the late start of glacier studies in the QTP, many efforts are still at the stage of collecting basic
data, such as the topographic and geomorphological information, as well as long-term field observations of ice
thickness, length and storage change in individual locations. A scientific group in the Institute of Cold and Arid
Regions Environmental and Engineering Research of the Chinese Academy of Sciences has conducted a series of
experiments and confirmed that glaciers at the northward aspect of the Himalayan Mountains trace monsoon
changes over a long historical period (Ma et al., 2010). Another important finding is that small glaciers tend to be
thinner when they span a greater vertical range because a greater vertical range is associated with greater slopes,
velocities, and driving stresses (Haeberli and Hoelzle, 1995).
Studies of the relationships between the thickness, area and volume of different glaciers are currently mainly based
on empirical parameters. For instance, Erasov (1968) described the relationship between area ($A$) and volume ($V$)
as $V=0.027 \cdot A^{1.5}$ for glaciers in central Asia. The Lanzhou Institute of Glaciology and Geocryology in China
(LIGG) (1986) defined the empirical relation between the glacier area and volume for the glaciers in western China
as $V=H \cdot A/1000$, and $H=53.2 \cdot A^{0.3}-11.3$ (where $H$ is the ice thickness). This equation was created to estimate the
glacier volume for a large region with numerous glaciers in China. Machereet et al. (1988) indicated the
relationship between the area and volume of the glaciers in the Altai-Tien Mountains as $V=0.0298 \cdot A^{1.379}$. Liu et
al. (2003) proposed the equation $V=0.0395 \cdot \cot (A^{1.35})$" for glaciers in the Qilian and Tien Mountains in Northwest
China.



However, the vertical extent of a glacier mostly spans a larger range of climatic conditions with a greater mass
balance difference from top to bottom. As a result, the flow at the equilibrium line is greater, which dominates for
larger glaciers (Grinsted, 2013). This point challenges the accuracy of volumes determined using the area-volume
scaling law on which the equations presented above are based. Hence, more field measurements must be collected
and new methods must be explored to obtain more accurate estimates of glacier volumes. With the development
of technology, field altimetry technology, such as airborne radio-echo sounding tracks, has been widely used. For
instance, one of the bedrock topography products was provided by CReSIS, University of Kansas and NASA
Operation Ice Bridge (https://data.cresis.ku.edu/). The Greenland Ice Mapping Project (GIMP) also employed this
technology and published the surface elevation measurement data (Howat et al., 2014). Subsequently, the Ice
Thickness Models Intercomparison Experiment (Farinotti et al., 2017; ITMIX) assessed the ability of seventeen
different approaches to reproduce the observed thickness for various glacier types around the globe. An outstanding
approach among these techniques is the ground-penetrating radar (GPR), which is considered to possess a strong
penetration function (Sun et al., 2002). GPR has been widely used to detect ice thickness (Wang and Pu, 2009;
Wu et al., 2011), subglacial topography (Zhu et al., 2014), and glacial hydrology features in recent years. The
combination of GIS, GPS and GPR provides access to knowledge of the ice thickness and volume distribution (Ma
et al., 2010). In China, GPR has been implemented in many cold areas for glacier monitoring since the 1980s
(Wang and Pu, 2009; Ma et al., 2010; Zhu et al., 2014; Huai et al., 2015).
**1.3 Purpose of this study**
Given the importance of the QTP in global water systems and climate systems, as well as the trend of glacier
melting amid global warming, complete databases/inventories are needed to record the glacier status and changes
over the years. The glacier volume data are essential for glacier-related studies, particularly for understanding the
effects of climate change. However, the complicated topographic and geomorphological conditions, and harsh
weather in the glacial area pose substantial challenges to the monitoring projects. The implementation of field
monitoring would not only require efficient technologies, but also large labour and financial resources. The
existing field observations are extremely scattered and very scarce. Therefore, a tool that compiles glacier
inventories based on the available remote sensing products and an appropriate calculation algorithm are
necessary. Currently, several glacier inventories have been complied. The Randolph Glacier Inventory (RGI, it
has version 1.0, 2.0, 3.2, 4.0, 5.0 and 6.0), and the First and Second Glacier Inventory of China (written as GIC-I
and GIC-II, respectively, below) are the most comprehensive inventories covering the QTP. The information
contains the minimum, median and maximum elevations, central location, mean slope, aspect, and area for each
glacier. Many aspects of data from glaciers in the Chinese territory included in the Randolph Glacier Inventory
have been improved based on the GIC-I (but the original GIC-I inventory is not available online). Meanwhile,
the RGI 4.0, 5.0 and 6.0 have been improved substantially compared to RGI 1.0, 2.0, and 3.0. In terms of the
data source dates over the QTP, 84.34% of images were collected from 1956~1980 in RGI 4.0, while all source
maps in the RGI 5.0 and RGI 6.0 were obtained from 1998~2010. GIC-Ⅱ includes the glacier data representing
the situation since 2000.
Glacier outline maps in different periods are required to study glacier evolution under the changing climate
conditions. The two inventories, RGI 4.0 and GIC-II, provide the opportunity to investigate the effects of climate
change on glaciers in QTP in the past few decades. However, RGI 4.0 did not provide information on glacier





volumes, while the data in GIC-II contain some overestimations/underestimations compared with the observed
data. Meanwhile, the mean thickness of the glaciers is not provided in either inventory. These gaps must be filled
and the existing data in the two inventories must be verified to provide robust databases for glacier-related
studies.
This study has two aims. The first is to recover the individual glacier volumes over the QTP based on the
existing glacier information in RGI 4.0 and GIC-Ⅱ. A slope-dependent algorithm (the specific description is
provided in Section 4.1) is applied for the calculation. The recalculated glacier volumes will be validated with
the data from published studies and field observations. The second aim is to investigate the effects of climate
change impacts by comparing the two glacier inventories, which represent the statuses at different periods. The
results can provide a basis for understanding the glacier evolution in the QTP in the context of climate change.
Moreover, the comparison would be helpful to capture the association between glacial retreat or advance with
different atmospheric circulation patterns, which will enable a re-tracking the signal of historical climate change
and project the changes into the future.
**2 Study area**
**2.1 Topographical and geomorphological characteristics**
The QTP is located in western China and surrounded by a large number of huge mountains (Figure 1), including
southern Himalayan, northern Qilian, Kunlun, western Karakorum, eastern Hengduan, and interior Tangula,
Gandise, and Nyainqentanglha Mountains. The majority of mountains extend from northwest to southeast. Most
have a height greater than 6000 m a.s.l. (above sea level), whereas the elevation at many mountain peaks in the
Himalayas even exceeds 8000 m a.s.l. In general, the average elevation over the entire QTP with total area of
approximately 2.5 million km$^2$ is greater than 4000 m a.s.l. Thus, the QTP has two nicknames: "the Third Pole"
of the earth and "The Water Tower of Asia".
The unique geomorphology of the QTP has largely resulted in the boundary discrepancy, with high mountains
located at the southwestern border and deep cuts located at the eastern margin. Due to the block of high
topography at southwestern border, water vapour from the Indian Summer Monsoon (the main source of water
vapour) is largely prevented from reaching the interior of the QTP. Only the area at southeastern Plateau with a
large water vapour channel intercepts a large amount of precipitation (the annual precipitation exceeds 4000
mm).

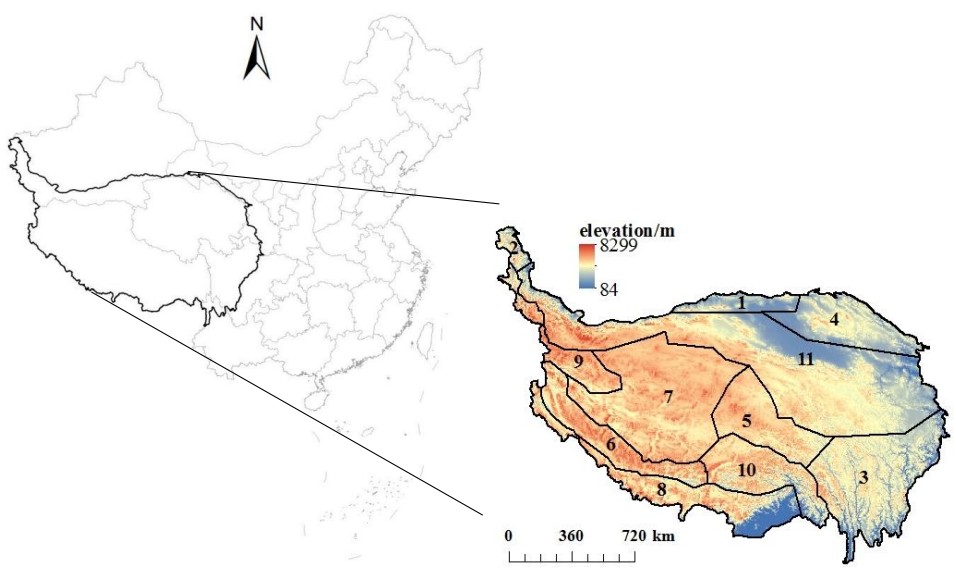

**Fig. 1 Location and surface elevation pattern of the QTP in China**

Note: 1-Altin Mountains (length: 730 km; width: 100 km); 2-Pamir Plateau (area: $10^5$ km$^2$; length: 260 km; width: 50-100 km); 3-Hengduan Mountains (area: $6 \times 10^5$ km$^2$; length: 900 km); 4-Qilian Mountains (length: 800 km; width: 200-400 km); 5-Tangula Mountains (length: 700 km; width: 150 km); 6-Gandise Mountains (length: 1100 km; width: 60-100 km); 7-Qiangtang Plateau (area: $5.97 \times 10^5$ km$^2$; length: 1200 km; width: 760 km); 8-Himalayan Mountains (length: 2450 km; width: 200-350 km); 9-Karakoram Mountains (length: 800 km; width: 240 km); 10-Nyainqentanglha Mountains (length: 1400 km; width: 80 km); and 11-Kunlun Mountains (area: $5 \times 10^5$ km$^2$; length: 2500 km; width: 130-200 km) (Guo, 2011).

**2.2 Glaciers and climate change**

Glacier changes in the QTP are largely attributed to the changing regional water vapour and energy conditions (Deng and Zhang, 2018; Qiu, 2008). The sources of water vapour over this region mainly include the Indian Summer Monsoon, westerlies and East Asia Monsoon (Moor and Stoffel, 2013). In the context of global climate change, these climate systems are altered, causing changes in the glaciers located in the QTP. Due to its complicated topography and geomorphology, and monsoon-surrounded atmospheric circulation conditions, regional warming over the QTP is quite substantial and three times higher than other areas in China (Qiu, 2008; Yao et al., 2012a). The warming climate induces glacier melting.

The QTP has a typical plateau climate with low temperatures and strong solar radiation (Luo et al., 2004). The isotherm in the QTP is rising from the northeast and eastern borders to the southwestern area, with the lowest isotherm in the Qilian Mountains and the eastern margin of the plateau and the highest isotherm in the southwestern plateau (Yao and Zhang, 2015). The distribution of annual precipitation in the QTP shows a decreasing trend from southeastern to northwestern areas (Qi et al., 2013). In general, the climate in the QTP presents a pattern of warm-wet in the southeast and dry-cold in the northwest (Wang et al., 2002).



**3 Input Data**
**3.1 Randolph Glacier Inventory version 4.0 (RGI 4.0)**
The Randolph Glacier Inventory 4.0 (RGI 4.0, http://www.glims.org/RGI/randolph40.html, doi:10.7265/N5-
RGI-40)) (RGI Consortium, 2014) was released on 1 December 2014 by Global Land Ice Measurements from
Space (GLIMS), which is a project designed to sketch glacier outlines all over the world based on the database
obtained from optical satellite instruments (Raup et al., 2007). The RGI 4.0 includes the glacier information on
central location, area, mean slope, mean aspect, maximum elevation, median elevation and minimum elevation
for each glacier. The glaciers in the Chinese territory in the RGI 4.0 were compared to the first Glacier Inventory
of China (GIC-I) based on topographic maps, aerial photographs and field measurements conducted in 1950s-
1980s (Shi et al., 2008, 2009; Wu & Li, 2004). Some empirical observations reported in scientific publications
were used to further validate the Chinese glacier data in the RGI 4.0, including the glacier inventory in the
Nyainqentanglha Range of southeastern Tibet from Bolch et al. (2010), a glacier layer from the Digital Chart of
the World (DCW) and the World Glacier Inventory (Raup et al., 2000; Haeberli et al., 1989; Haeberli et al.,
1998). Most glacier outlines in the central and eastern Himalayas and Karakoram Mountains were obtained from
the project of International Centre for Integrated Mountain Development (ICIMOD) (Bhambri et al., 2013; Frey
et al., 2012; Mool et al., 2007; Raup et al., 2007), glacier outlines of the northeastern Karakoram Mountains were
obtained from the study by Bhambri et al. (2013), and most data on the northern slopes of the Himalayas and the
northeastern part of the Karakoram Mountains were obtained from the GIC-I (Shi et al., 2009).
**3.2 The second Glacier Inventory of China (GIC-II)**
The basic information for each glacier in the GIC-II (http://westdc.westgis.ac.cn,
doi:10.3972/glacier.001.2013.db) (Guo et al., 2014) is same as the RGI 4.0. The GIC- II reported the above-
mentioned glacier properties during 2007-2012 based on 218 Landsat images (http://earthexplorer.usgs.gov/), in
which the widely-used band-ratio segmentation and manual adjustment were applied to outline glaciers.
Meanwhile, several high-resolution images and Global Positioning System (GPS) measurements were combined
to validate the results of glacier delineation. Delineations of the ice divide were based on DEMs (cell size of
30m) generated from digitized topographic maps, which were mainly constructed from aerial photographs
acquired during the 1950s-1980s. In addition, two types of digital elevation models (DEMs) were used during
the compilation of GIC-II to acquire altitudinal range of individual glaciers. A seven-coefficient transformation
was employed on the elevation points and the digitized contours before DEM generation in order to minimize
potential errors introduced by the mismatch in different coordinate systems, like Landsat images and topographic
maps. The coefficients were obtained from coordinates of national trigonometric stations within and around
maps collected from the Mapping and Geoinformation of China, the National Administration of Surveying. In
the process, the Shuttle Radar Topographic Mission (SRTM) DEM from the Consultative Group for
International Agriculture Research (CGIAR) version 4, where voids were filled using different auxiliary DEMs
(http://srtm.csi.cgiar.org), were used to derive topographic attributes of the glaciers (Guo et al., 2014, 2015).
**3.3 Bedrock elevation map and Digital Elevation Model outputs**
ETOPO1 Global Relief Model was built using GMT 4.3.1 (http://gmt.soest.hawaii.edu/). GMT 4.3.1 creates
grids with the spatial resolution of 1 arc-minute (625 m) in a netCDF COARDS-compliant format. The grid of



the Earth's surface successfully depicts the bedrock underneath the ice sheets using ETOPO1
(https://www.ngdc.noaa.gov/mgg/global/global.html, doi:10.7289/V5C8276M) (Soller and Garrity, 2018). The
bedrock elevation dataset was obtained by using the MB-System (http://www.ldeo.columbia.edu/res/pi/MB-
System/) based on sixteen datasets, including Antarctica RAMP Topography, Antarctica BEDMAP Bedrock,
Greenland NSIDC Bedrock, Gulf of California Bathymetry, Mediterranean Sea Bathymetry, JODC Bathymetry,
Baltic Sea Bathymetry, IBCAO Bathymetry, Caspian Sea Bathymetry, U. S. Coastal Relief Model, Great Lakes
Bathymetry, Created Iceland Bathymetric Surface, SRTM30 Global Topography, GLOBE Topography,
Measured and Estimated Seafloor Topography, Bathymetric pre-surface. In this process, the "mbgrid" gridding
algorithm, a tight spline tension to the xyz data, based on the data hierarchy was utilized to interpolate values for
cells without data. The data hierarchy follows the relative gridding weights, in which the Antarctica RAMP ice
surface topography, Antarctica BEDMAP bedrock topography, the Greenland NSIDC bedrock topography
datasets were given the greatest weight (Amante and Eakins, 2009). Considering the bedrock elevation data over
the QTP were completely interpolated by the above-mentioned algorithm, relevant results from previous studies
explained the availability of this dataset in the QTP (Thompson et al., 1989, 1990, 1995; Liu et al., 1998; Li et
al., 2011). Most upland areas are composed of exposed bedrock and patchy glacier deposits. In these regions, the
land and bedrock topography are in close proximity. Thicker glacier deposits are largely located in lowland
areas, for which little or even no relation exists between the bedrock and land-surface topography. In grids in
which the calculated bedrock elevation exceeded the surface elevation, the latter is substituted by the land-
surface values (Amante and Eakins, 2009).
Shuttle Radar Topography Mission (SRTM) output on grids with a spatial resolution of 30 m (SRTM DEM 30
m) in 2001 (https://dds.cr.usgs.gov/srtm/version2_1/SRTM30/) over the QTP was collected to determine the
location of the grid cell with the maximum surface elevation for individual glaciers included in RGI 4.0. In
practice, the SRTM DEM 30 m is first transformed into the grids of the bedrock elevation data with the spatial
resolution of 625 m by using the resampling tool based on the nearest technique to match the bedrock elevation
data. The specific usage is described in Section 4. Moreover, the slope and aspect data are produced by the
elevation obtained from the SRTM DEM 30 m map through the ArcGIS platform. The slope data identify the
rate of maximum change in elevation from each grid. The aspect, the slope direction, captures the downslope
direction of the maximum rate of change in elevation from each grid to its neighbors. In the usage, the extracted
slope data were considered to keep consistent in the glacier surface during the study period. On the one hand,
most glaciers in the Qinghai-Tibetan Plateau are mountain glaciers. For those located in high slopes, the surface
slopes and corresponding bottom slopes are in close proximity. Therefore, even the glaciers move, the slope data
hardly change (Aizen et al., 2002). While in terms of glaciers at smaller slopes, the movement of glaciers is
slighter (Lambrecht et al., 2011). Considering the neighboring grids tend to be in a similar climate condition, the
changes of surface elevations among these grids are in a significant synchronism when the glaciers melt (Vieli
Leysinger and Gudmundsson, 2010). In general, the slope data are relatively stable. It is available to infer the
glacier surface elevation distribution with them.
**3.4 Glacier thickness data for validation**
The World Glacier Monitoring Service (WGMS) (https://wgms.ch/) is a global program devoted to collecting
and mapping glacier inventory datasets worldwide. The subset Glacier Thickness Dataset version 2.0 (GlaThiDa,
http://dx.doi.org/10.5904/wgms-glathida-2016-07, doi:10.5904/wgms-glathida-2016-07) (WGMS, 2016) stores
several glacier thickness measurements collected from field observations worldwide. The dataset has been
structured into three data tables. The first table is the overview table (T-GLACIER THICKNESS OVERVIEW)
and contains information on the location and area of the glacier, estimates of thicknesses from interpolated
observations. The second table (TT-GLACIER THICKNESS DATA DERIVED FROM MAP or DEM) includes
ice thickness data (mean and/or max) averaged over the surface elevation bands established based on the lower
and upper boundaries from ice thickness maps or Digital Elevation Models (DEMs). The third table (TTT-
GLACIER THICKNESS POINT DATA) contains point data including the elevation at the surveyed point, and
the thickness value (Gärtner-Roer et al., 2014). This dataset was applied to validate in the calculated glacier
thickness in the present study.
**3.5 Glacier volume data for validation**
The calculated glacier volumes were validated using glacier volume and change data obtained from the literature
and observations. The specific information retrieved the data in the literature is listed in Table A1 (Appendix). In
addition, the derivations of gravity anomaly (DGA) data over the QTP
(http://www.geodoi.ac.cn/WebCn/doi.aspx?Id=539, doi:10.3974/geodb.2016.06.07.V1) (Liu et al., 2016) provide
a sum of changes in soil moisture and glacier volume from 2003 to 2010 on the grids with spatial resolution of 1°,
which were sourced from Gravity Recovery and Climate Experiment outputs (GRACE) (Liu et al., 2015, 2016).
Soil moisture data with spatial resolution of 0.25° were extracted from the Global Land Data Assimilation System
(GLDAS) products (https://ldas.gsfc.nasa.gov/gldas/, doi:10.5067/LYHA9088MFWQ) (Hiroko and Rodell, 2016)
during the same period to obtain the changes in glacier volumes included in the DGA dataset. Based on these
datasets, the change in glacier volume in 1°×1° pixels is calculated by subtracting the DGA value from the
corresponding GLDAS soil moisture value (resampled from the 0.25°×0.25° to the 1°×1° pixel). Moreover, a
newly generated estimation of global glacier ice thickness data produced by Farinotti et al. (2019) was used as an
ancillary validation dataset, which was downloaded at the website "https://www.research-
collection.ethz.ch/handle/20.500.11850/315707" (doi:10.3929/ethz-b-000315707). The data were developed
based on the RGI 6.0, which is consistent with the GIC-II over the QTP. Thus, the glaciers with observed thickness
data have also been selected from the dataset reported by Farinotti et al. (2019) and compared with the recalculated
and traditional equation-based ice thickness. All of the above-mentioned data collections are listed in Table 1.

273                   **Table 1 General information about the data collections**

| Data types | Datasets | Usages |
| --- | --- | --- |
| Input data | Randolph Glacier Inventory version 4.0 (RGI 4.0, http://www.glims.org/RGI/) | Basic glacier property data (outlines, slope, aspects) |
| | The second Glacier Inventory of China (GIC-II) | |
| | Surface elevation data (SRTM DEM, 2001) | Ice thickness and volume calculation |
| | Bedrock elevation map | |
| Validation data | Glacier Thickness dataset version 2.0 (https://wgms.ch/) | Ice thickness and volume data in different space scales for validation |
| | Glacier volume data from the literature (Ma et al., 2008; Wang & Pu, 2009; Gärtner-Roer et al., 2014; Zhu et al., 2014) | |
| | Derivations of Glacier Anomaly (DGA) data (http://data.tpdc.ac.cn/), | |



| Global Land Data Assimilation System (GLDAS) dataset |
| Estimation of glacier ice thickness data by Farinotti et al. |
| (2019) (https://doi.org/10.3929/ethz-b-000315707) |

**4 Methods for calculating glacier volume**
**4.1 Ice thickness and volume determination**
Because the majority of glacier data over the QTP in RGI 4.0 are obtained from the GIC-I, the recommended
equation in the GIC-I is assumed to be applied to the RGI 4.0 data. In both GIC-I and GIC-II, the volume-area
scaling law was based to calculate the individual glacier volume (Wu & Li, 2004; Guo et al., 2015). The following
specific equations are recommended:
$$V_{\mathrm{I}} = \begin{cases} 0.0305 \cdot A^{1.11}, & A < 1\,\mathrm{km}^2 \\ 0.0542 \cdot A^{1.06}, & 1\,\mathrm{km}^2 \le A \le 3\,\mathrm{km}^2 \\ 0.0674 \cdot A^{1.16}, & A > 3\,\mathrm{km}^2 \end{cases} \tag{1}$$

$$V_{\mathrm{II}} = \begin{cases} 0.0365 \cdot A^{1.375}, & \text{maximum approximation} \\ 0.0433 \cdot A^{1.29}, & \text{minimum approximation} \end{cases} \tag{2}$$

where $V_{\mathrm{I}}$ and $V_{\mathrm{II}}$ represent the glacier volume in GIC-I and GIC-II, respectively. The units of $A$ and $V$ are km² and
km³, respectively. $V_{\mathrm{I(II)}}/A$ is applied to calculate the average thickness of a glacier.
As mentioned above, data are missing from the two inventories and problems of over and underestimations of
glacier volumes have been noted (Guo et al., 2015). The aforementioned datasets and the derived slope and
aspect maps over in the QTP were applied to recover the missing data, determine ice thickness, and recalculate
glacier volumes in the two inventories as a method to improve the accuracy of the calculations.
In addition, there is an assumption that the grid with the maximum surface elevation in an individual glacier is
assumed to remain unchanged during two studied periods (Erasov, 1968; Gardelle et al., 2013; Frey et al., 2014).
The following specific procedures were used:
1) Select the grid location ($x_0$, $y_0$) with the maximum elevation ($Z_0$) of glaciers in the RGI 4.0 using the
surface elevation map constructed in 2001 (SRTM30) in the QTP.
2) Use the following slope-dependent algorithm shown below (Fig. 2) to obtain the surface elevation map
based on the identified maximum elevation location (grid) identified in step 1.
① Identify the adjacent pixels of the grid with the maximum elevation recognized in step 1 (the distance
between centres of two grids' at a spatial resolution of 1km is equal to or less than 1.45 km (the largest centre
distance between two neighbouring pixels). These pixels are labelled as $i$, $i=1$ 2, ..., $n_1$, with a surface elevation
$Z_{1,i}$, and location ($x_{1,i}$, $y_{1,i}$)), and then calculate the surface elevation for these pixels.
$$Z_{1,i} = Z_0 - \tan(mean(slope_0 + slope_{1,i})) \times sqrt((x_{1,i} - x_0)\wedge 2 + (y_{1,i} - y_0)\wedge 2) \tag{3}$$

② Identify the adjacent pixels of the grid $i$ identified in ①; (the found pixels are designated as $j$, $j=1, 2, ...,$
$n_2$, with surface elevation $Z_{2,j}$, and location ($x_{2,j}$, $y_{2,j}$)).
$$Z_{2,j} = Z_{1,i} \pm \tan(mean(slope_{1,i} + slope_{2,j})) \times sqrt((x_{1,i} - x_{2,j})\wedge 2 + (y_{1,i} - y_{2,j})\wedge 2) \tag{4}$$
$$\pm : depending\ on\ the\ aspect\ comparison\ of\ grid\ i\ and\ j$$

③ Repeat ② from $i=1$ to $i=n_1$ until the boundary pixels are identified.



3) Calculate grid-based ice thickness ($H$) in combination with bedrock elevation map (the 1km×1km grid
was used for the calculation).

$$H = Z_{2,j} - Z_{B,j} \qquad (5)$$

where $Z_{B,j}$ is the bedrock elevation corresponding to $Z_{2,j}$.
4) Based on the grid-based ice thickness, the individual glacier volume was computed using the following
equation:

$$V = \bar{H} \times A \qquad (6)$$

where $\bar{H}$ is the pixel-averaged ice thickness.
In this process, the maximum surface elevation grid is synchronous to the grid cell of the maximum bedrock
elevation (according to the correlation analysis, the correlation coefficient between the two series is greater than

0.85).

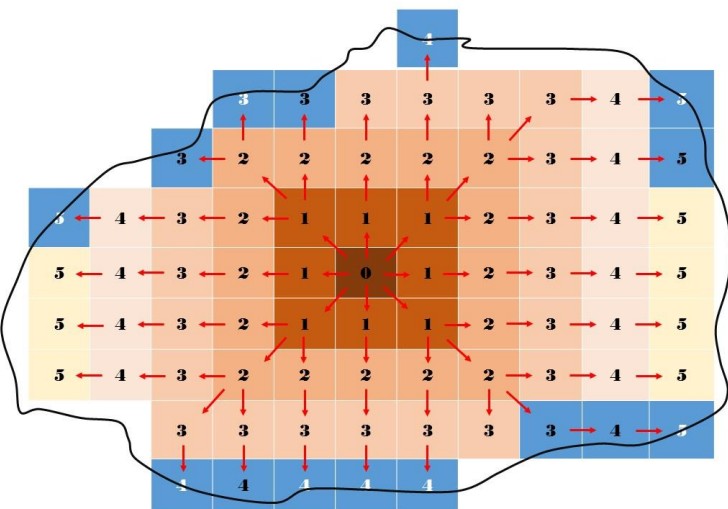


**Fig. 2 Schematic map for glacier volume calculation**

Note: The numbers 0-5 indicate the surface elevation of the pixels. Grids with the same number indicate a
contour line.
**4.2 Fragmentation index**
According to previous studies, total glacier numbers have increased in recent decades, although glacier areas are
significantly decreasing. The fragmentation index introduced in landscape-related studies was adopted and
computed using the following equation to analyse the changes in glacier numbers in different areas during the past
few decades:

$$FI_i = \frac{N_{\text{GIC-II},i} / N_{\text{RGI 4.0},i} - 1}{(\sum_{t=1}^{N_{\text{RGI4.0}}} A_{\text{RGI4.0},i,t} / \min(A_{\text{RGI 4.0},i})) / (\sum_{t=1}^{N_{\text{GIC-II}}} A_{\text{GIC-II},i,t} / \min(A_{\text{GIC-II},i}))} \qquad (7)$$


where $i$ ($i=1, 2, 3, ..., 11$) represents the code for different mountains. $FI_i$ is the fragmentation index of $i$. $N_{\text{RGI 4.0},i}$





and $N_{\text{GIC-II},i}$ refer to the glacier number of mountain $i$ in RGI 4.0 and GIC-II, respectively. $A_{\text{RGI 4.0},i,t}$, $A_{\text{GIC-II},i,t}$ are
the area of the glacier $t$ in mountain $i$ in RGI 4.0 and GIC-II, respectively. $\min(A_{\text{RGI 4.0},i})$ and $\min(A_{\text{GIC-II},i})$ mean
the minimum glacier area in mountain $i$ in RGI 4.0 and GIC-II, respectively.
A higher fragmentation index indicates that more surfaces are exposed to sunlight, which might result in more
energy accepted by glaciers to produce more meltwater. Meanwhile, the shear stress would also increase and
basal sliding would accelerate, which is the key interpretation of how the glacier movement and deformation will
develop.
In addition, the ratio of disintegrated glaciers (RDG) is computed as follows.
$$\text{RDG} = \frac{\text{GIC-II\_GN} - \text{RGI 4.0\_GN} + \text{Disappeared\_GN} - \text{Surged\_GN}}{\text{RGI 4.0\_GN}} \qquad (8)$$

where RGI 4.0_GN, GIC-II_GN are the glacier number in RGI 4.0 and GIC-II, respectively. Disappeared_GN and
Surged_GN indicate the number of disappeared and surged glacier number from the 1970s to the 2000s,
respectively.
The following equation was used to calculate the average number of glaciers in the GIC-II that disintegrated from
a glacier in the RGI 4.0.
$$\text{DGN} = \frac{\text{GIC-II\_GN} - \text{Surged\_GN}}{\text{RGI 4.0\_GN} - \text{Disappeared\_GN}} \qquad (9)$$

where DGN presents the glacier number that disintegrated from the RGI 4.0 to GIC-II.

**4.3 Uncertainty estimation**

According to a large amount of statistics, approximately half area of boundary pixels are included in the glacier.
In the practical calculation, the whole area of boundary pixels is counted in the glacier volume. Thus, the product
of the number of boundary pixels and half area of each pixel is computed to quantify the uncertainty in the glacier
area using the following equation (Shi et al., 2009; Guo et al., 2015).
$$\varepsilon = N \cdot A' \qquad (10)$$

where $N$ is the number of boundary pixels and $A'$ is the half area of each pixel. The individual glacier volume is
finalized as the range of $(A \pm \varepsilon) \cdot H$ to include the uncertainty in the glacier area.

**5 Results**

**5.1 Validation of the calculated ice thickness and volume**

Using the input data and the methods specified above, the ice thickness and volume of individual glaciers are
calculated. The calculated values for selected glaciers are compared with the observed data and corresponding
equation-based results in RGI_4.0 and GIC_II (Fig. 3). Most of the calculated glacier volumes display better
agreement with observations than the equation-based results for the selected glaciers, particularly in the
Nyainqentanglha Mountains (Bayi and Gurenhekou Glaciers). While at the Shule_5 and Shule_6 glaciers in the
northern Qilian Mountains, both calculated and equation-based thickness and volume values are much larger than
the observed values in the RGI 4.0. In the GIC-II, Farinotti et al.'s (2019) results tend to be lower than the other
two values. In general, errors in the provided slope-dependent algorithm are smaller than errors for the equation.
Therefore, it will be further used to calculate all individual glacier volumes in RGI 4.0 and GIC-II. The relevant
descriptions are provided below.

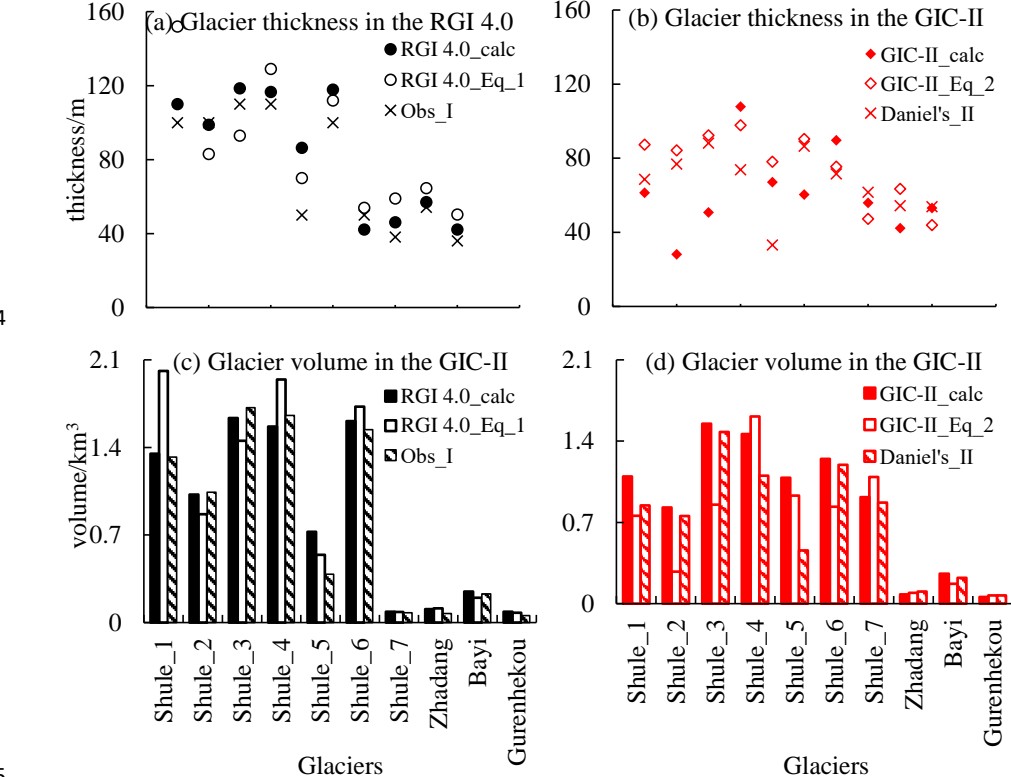

**Fig. 3 Comparisons between observed glacier volumes and values calculated using different methods**

Note: The ice thickness values are reported as average values. RGI 4.0_calc, RGI 4.0_Eq_1, and Obs_I are the calculated, Eq. (1)-based, and observed values in the RGI 4.0, respectively. GIC-Ⅱ_calc, GIC-Ⅱ_Eq_2 and Daniel's_Ⅱ represent the calculated, Eq. (2)-based values (the averages of the minimum and maximum values), and Farinotti et al.'s (2019) results obtained by averaging five types of glacier model outputs in the GIC-Ⅱ.

Three GRACE data grids with a spatial resolution of 1° (approximately 100 km×100 km) from the Himalayan Mountains are also chosen to further compare and validate the calculated results and products of glacier volume change as shown in Table 2. An underestimation is observed in the results obtained with the volume-area scaling. In particular, the approximately 45.6%~58.4% rate of change in the total glacier volume has been underestimated by Eq. (2) during 2003-2009 on the west-most grid, but an underestimation of only 10.4% in the change in the total glacier volume occurs using the slope-dependent method. Moreover, a large extent of change in the glacier volume is given by the empirical equation for the central pixel selected, and, fortunately, the observed result is similar to the peak value of the range. In the eastern pixel, a ratio of 16.8-22.4% of change in the observed glacier volume change has been identified from the results of the volume-area scaling, while the calculation produces an overestimation of 6.5% in the DGA-derived change in the glacier volume.




**Table 2 Changes in glacier volume during 2003-2009/2010 in the selected DGA data grids**

| Central-lon | Central-lat | Period | Calculated glacier volume change km³ | Equation-based glacier volume change km³ | DGA-derived glacier volume change km³ |
|---|---|---|---|---|---|
| 86 | 28 | 2003-2009 | 22.4 | 10.4~13.6 | 25 |
| 87 | 28 | 2003-2010 | 8 | 1.9~8.9 | 8 |
| 88 | 28 | 2003-2010 | 11.4 | 8.3~8.9 | 10.7 |


The calculated results are also compared with relevant studies in the QTP presented in the literature (Table 3). All
selections have similar study periods with the corresponding information in the RGI 4.0 and GIC-Ⅱ, from which
the change in the glacier volume in the Gongga Mountains located in the Hengduan Mountains is overestimated
by the volume-area scaling because of the significant underestimation of the glacier volume in the GIC-Ⅱ.
Meanwhile, changes in the glacier volume in the central Nyainqentanglha Mountains and Dongkemadi Glacier in
the Tangula Mountains are underestimated by the traditional method due to the overestimation of glacier volume
in the GIC-Ⅱ. In addition, the results for the Laohugou No.12 Glacier in the Qilian Mountains calculated using
the slope-dependent and volume-area scaling are consistent mainly because the equation was determined based on
the observations over the surrounding area. In general, the comparison of the results reveals good agreement with
the verifications conducted above.

**Table 3 Comparison with changes in the observed glacier volume reported in the literature**

| Region | Period | Location/code | Observed volume change in references (km³) | Calculated volume change (km³) | Equation-based volume change (km³) |
|---|---|---|---|---|---|
| Gongga Mountains | 1966-2015 | 29°-30° N, 101°-102°E | － 1.65 (Cao et al., 2019) | － 1.06 | － 5.99 |
| Central Nyainqentanglha Mountains | 1968-2000 | 30.15°-30.88° N, 94°-95.5° E | － 23.62 (Brun et al., 2017; Wu et al., 2019) | － 17.32 | － 8.23 |
| Dongkemadi Glacier | 1969-2000 | 5K443D0038 | － 1.17 (Li et al., 2012) | － 1.32 | － 0.09 |
| Laohugou No.12 Glacier | 1957-2007 | 5Y448D0012 | － 0.22 (Liu et al., 2018; Zhang et al., 2012) | － 0.28 | － 0.5 |


**5.2 Changes in volumes**
The individual glacier volume equals the pixel-averaged ice thickness multiplied by the area (Eq. (6)). The sums
of individual glacier volumes in the recalculated RGI 4.0 and GIC-II inventories are provided in Table 4. The total
area of glaciers in the QTP extracted from the RGI 4.0 is approximately 54874.79 km², and the area extracted from
the GIC-II is 43745.48 km², representing a decrease of 11129.31 km². The total glacier volume was reduced from



4716.76 km$^3$ in the RGI 4.0 to 4045.59 km$^3$ in the GIC-II. The results suggest a significant retreat and melting of
glaciers in the QTP since the 1970s.
The glacier volumes of the Tangula Mountains, Qiangtang Plateau, Karakoram and Kunlun Mountains in the
inland Tibetan Plateau are lower than volumes calculated with the volume-area scaling (Table 4). However, the
calculated volumes for the other mountains are larger than the equation-based values. In the compilation of Glacier
Inventory of China, the traditional empirical equations were determined by the survey and monitoring on glaciers
from northern mountains, particularly in the eastern Qilian Mountains where lack extensive deep valleys (Liu et
al., 2003; Wu & Li, 2004; Shangguan et al., 2010). Thus, the total glacier volume in the Qilian Mountains is
underestimated. However, the inland Tibetan Plateau is filled with more flatter lands. This difference may cause
the overestimation of glacier volume in the inland Tibetan Plateau using the empirical area-volume equations
determined without enough available field observations in this area. Weakening of the westerlies and the Indian
Summer Monsoon might be the dominant factors limiting glacier accumulation in recent decades (Yao et al.,
2012b). On the other hand, the geological structure of the southwestern mountains tends to be more complicated
with a large number of deep valleys due to several gigantic orogenic movements in history. The higher mountains
with deeper valleys store thicker glaciers, leading to the underestimation of glacier volume by the volume-area
scaling in these areas. The causation is that deep valleys tend to store thicker glaciers, but lack observations (Guo
et al., 2015). Thus, the empirical equations cannot capture the information for these areas so the underestimation
is always bound. Moreover, traditional empirical equations tend to underestimate glacier volumes for small glaciers
(an area of less than 1 km$^2$), as well as parts of large glaciers with thin layers (an area greater than 10 km$^2$ and
thickness less than 40 m). Thus, most of the equation-based glaciers volumes are median in scale, while small
glaciers and large glaciers with thin layer always have a lower proportion of the volume to area ratio (Klein et al.,
2014; Wang et al., 2019).

**Table 4 Comparison of the calculated and equation-based glacier volumes in different mountains**

| Mountains | Glacier volume in RGI 4.0 (km$^3$) | | Glacier volume in GIC-II (km$^3$) | | Changing rate of glacier volume based on calculations (%) |
| --- | --- | --- | --- | --- | --- |
| | Calculation | Equation 1 | Calculation | Equation 2 | |
| Altin | 31.69 | 22.02 | 29.83 | 15.36 | -5.86 |
| Pamir | 249.35 | 221.10 | 281.66 | 166.32 | +12.96 |
| Hengduan | 200.88 | 175.93 | 131.27 | 77.03 | -34.65 |
| Qilian | 185.64 | 140.58 | 135.90 | 84.48 | -26.79 |
| Tangula | 178.72 | 229.97 | 123.59 | 140.25 | -30.85 |
| Gandise | 197.07 | 164.75 | 96.01 | 55.79 | -51.28 |
| Qiangtang | 192.92 | 254.26 | 136.93 | 166.93 | -29.02 |
| Himalaya | 705.29 | 618.48 | 603.92 | 497.66 | -14.37 |
| Karakoram | 501.81 | 553.40 | 524.87 | 589.32 | +4.59 |
| Nyainqentanglha | 1065.71 | 981.16 | 937.65 | 859.36 | -12.02 |
| Kunlun | 1207.68 | 1260.06 | 1044.18 | 1117.96 | -13.54 |

Note: The results listed in "Equation 1" are obtained from Eq. (1). The results listed in "Equation 2" are based on
the averages of the minimum and maximum values calculated using Eq. (2).




The comparison of the recalculated RGI_4.0 and GCI_II indicates that the glacier areas of all mountains over the
QTP have decreased in the past 4 decades. The smallest percent reduction in glacier area is observed in the Altin
Mountains, with the value of approximately 8.26% of the total area. In the Gandise Mountains, the glacier area
over all aspects has decreased to 45.36% of its total area, which is the largest percent reduction in glacier area
among all the studied mountains. Meanwhile, the change in the glacier volume in the Gandise Mountains is
consistent with the change in glacier area, i.e., it also exhibits the largest percent decrease of －51.28% (Table 4).
However, the Tangula Mountains contain an area of expansion on the northwestern aspect, while the glacier area
increased on the southwestern aspect in the Pamir and Qiangtang Plateaus. In addition, the increase in the glacier
area in the other mountains is mainly located on the northern and northeastern aspects. The majority of advancing
glaciers are distributed in the northwestern Karakoram Mountains with higher rate than other mountains. The
results are consistent with the study by Liu et al.'s (2014).
**5.3 Disappeared and surged glaciers from 1970s to 2000s**
The glacier information in the calculated RGI 4.0 and GIC-II is compared to detect the disappeared and surged
glaciers with the aid of the ArcGIS toolbox (Fig. 4). The disappeared glaciers refer to glaciers that were included
in the RGI 4.0 but did not appear in the GIC-II. The surged glaciers include the glaciers that were emerging in the
GIC-II. The statistical analyses of the disappeared and surged glacier numbers and volumes over different
mountains are displayed in Fig. 5. The Karakoram and Kunlun Mountains have comparably larger numbers of
surged glaciers, with values of 1598 and 1329, respectively. However, the Gandise and Himalayan Mountains
contain the greatest numbers of disappeared glacier at 1405 and 1387, respectively. The results are consistent with
the study by Bhambri et al.'s (2017).

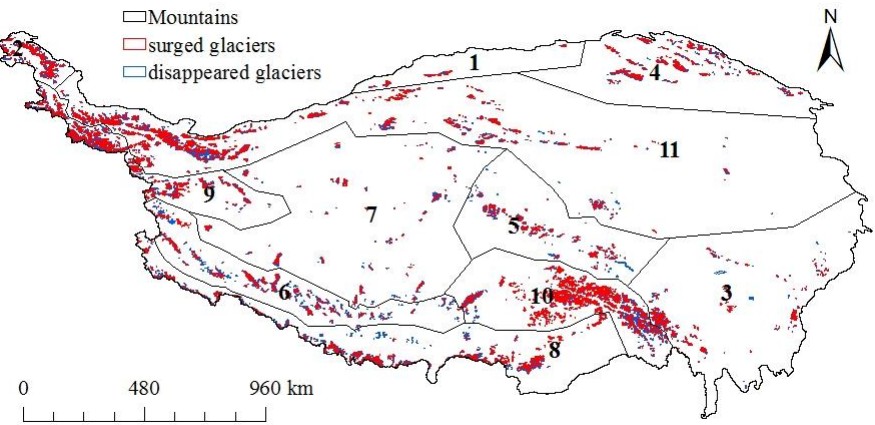


**Fig. 4 Disappeared and surged glaciers from the 1970s to 2000s over the QTP**

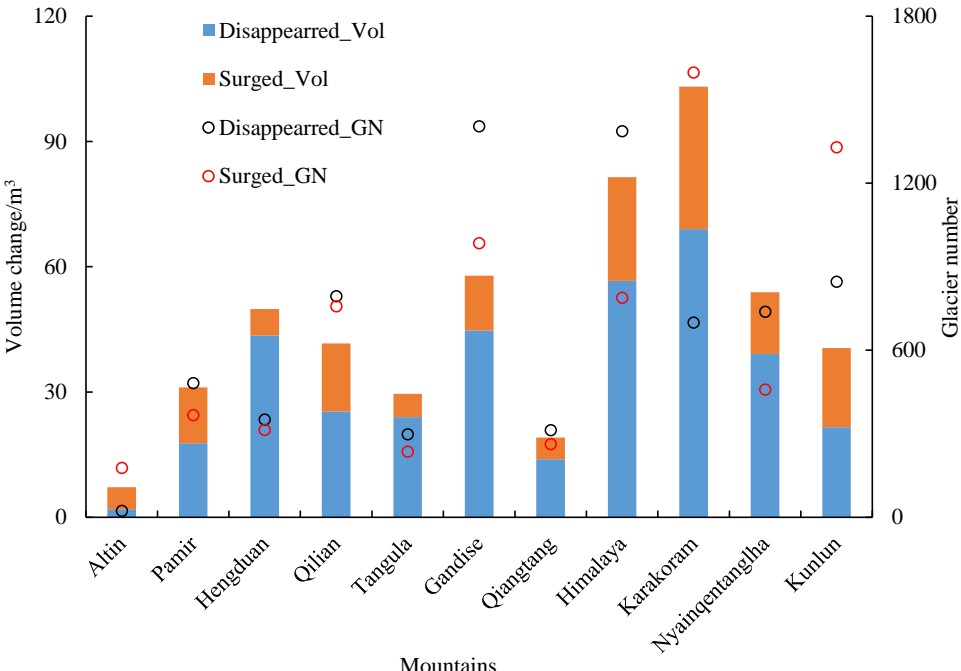


**Fig. 5 Statistical analyses of disappeared and surged glaciers in 1970s and 2000s in the QTP**

Note: Disappeared_Vol, Surged_Vol refer to the disappeared and surged glacier volume from the 1970s to the
2000s, respectively.

### 5.4 Effects of mountain directions on changes in glacier volumes in the RGI 4.0 to GIC-II

In the QTP, different mountains run in different directions (Fig. 5) and different aspects have different climatic
conditions, causing diverse glacier accumulation and ablation. In the context of climate change, the advance and
retreat of glaciers may vary in the different aspects of each mountain. Therefore, studies exploring changes in
glacier volume in different aspects are necessary to understand the mechanisms by which the changing climate
and monsoon affect the glaciers. The glacier volumes in the recalculated RGI 4.0 and GIC-II are summed for
different mountains to investigate the variations (Fig. 6). The changing pattern of glacier volume in the
Nyainqentanglha Mountains is similar to the Himalayas. A similar pattern of changes is also observed in the Qilian,
Kunlun and Gandise Mountains. The Altin Mountains have experienced a significant increase on the northern (0°-
15° and 345°-360°) aspect and slight increase on the eastern (75°-105°), south-southeastern (SSE, 135°-165°),
north-northwestern (NNW, 315°-345°) aspects. The Pamir Plateau displays a significant increase in glacier volume
on the east-southeastern (ESE, 105°-135°), north-northeastern (NNE, 15°-45°), west-southwestern (WSW, 225°-
255°) and west-northeastern (WNW, 285°-315°) aspects, while the glacier volumes decreased on other aspects. In
addition, the glacier volumes in the Hengduan Mountains and Himalayan Mountains have undergone an increase
in the eastward direction. The Qilian Mountains display an increase in glacier volume on the northern and NNE
aspect. The total glacier volume in the Karakoram Mountains increased on the northern aspect. The glacier volume
on the northern and NNE aspects underwent an increase in the Kunlun Mountains. Moreover, the Nyainqentanglha
Mountains exhibit an increase in glacier volume on the eastern aspect. The Qiangtang Plateau displays a slight
increase in the glacier volume on the east-northern aspect. However, the glacier volumes at all aspects in both the
Tangula and Gandise Mountains were reduced during the study period. On the other hand, the statistics of glacier
volume changes on different aspects described above reflect glacier movement.
In summary, the volume of glaciers in most of mountains decreased on the western and southern aspect in the
studied period. The aspects displaying a reduction in glacier volume in the northern mountains are concentrated
on the northern and northeastern aspect, while the southern mountains mainly exhibit a decrease in glacier volume
on the eastern and even a few southeastern aspects. Thus, the concentrations of aspects of the mountains with
increasing glacier volumes from north to south are shifting from north and northeast to east and southeast,
indicating that glacier retreat occurs in the southwestern aspects of mountains, while the northeastern aspects of
mountains tend to display glacier advance.

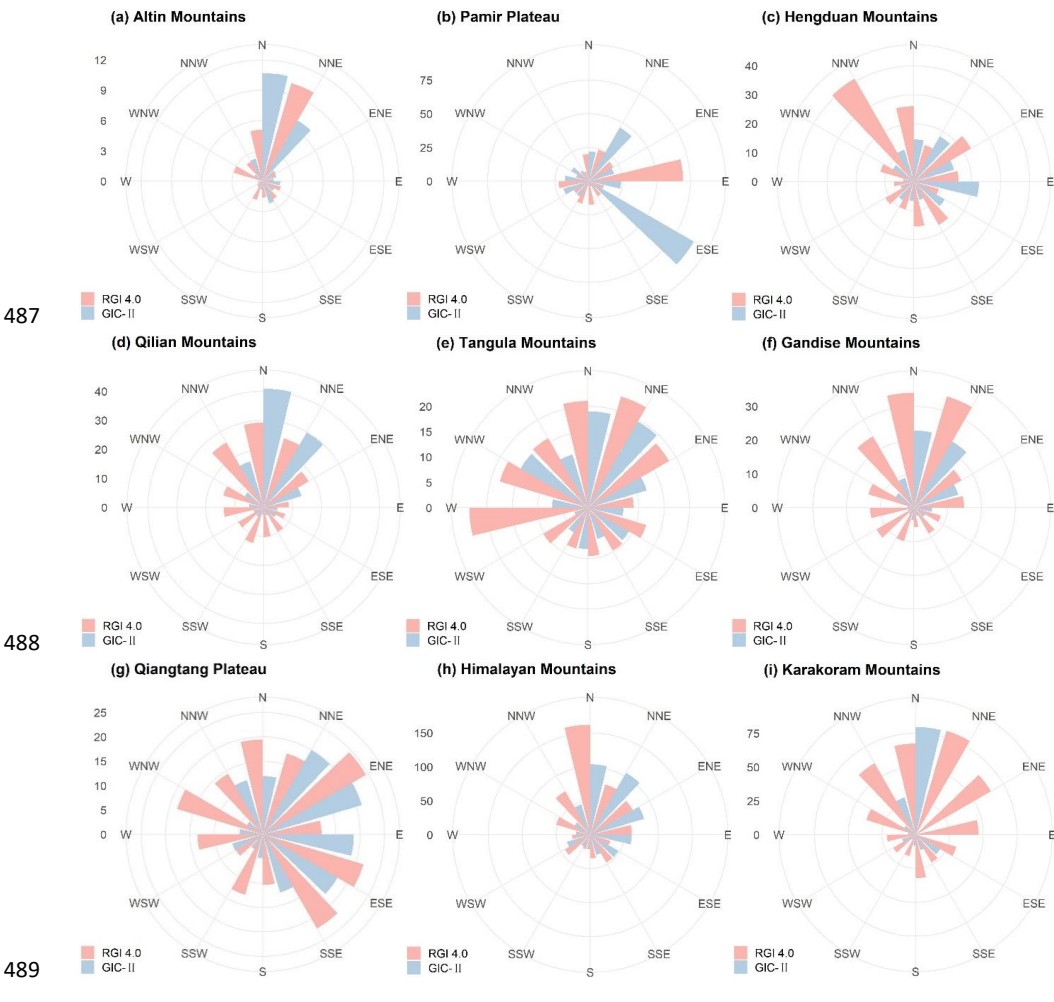




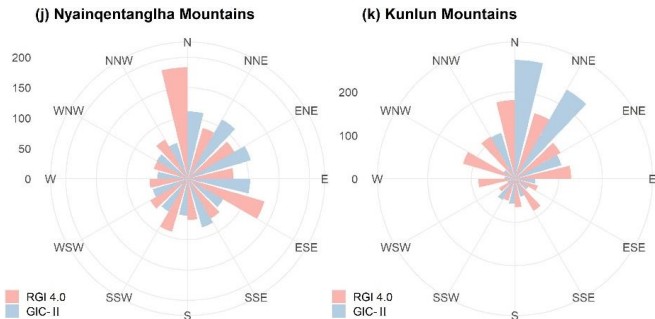


**Fig. 6 Glacier volumes on different aspects of eleven mountains in calculated RGI 4.0 and GIC-II** (unit:

km$^3$)

Note: The statistical analyses of glacier volumes in each mountain are conducted on twelve direction ranges

(anticlockwise is defined as positive direction; due north is 0°). For instance, the range of 15°-45° refers to the

north-northeast orientation (NNE). Areas of the fan-shaped sector coloured in red and blue represent the glacier

volumes in the calculated RGI 4.0 and GIC-II, respectively.

**5.5 Glacier fragmentation**

Glacier melting can lead to the disappearance of small glaciers and the fragmentation of a part of large glaciers

(Liu et al., 2014). To quantify such fragmentation, the fragmentation indexes of glaciers in different mountains

from RGI 4.0 to GIC-II are calculated using Eq. (7). The results are shown in Fig. 7. It is obvious the values of

fragmentation index are either positive or negative, which directly depend on the change of glacier number from

RGI 4.0 to GIC-II in different mountains. The larger the fragmentation index is, the greater glacier number in

GIC-II has, or the smaller decrease of glacier area from the RGI 4.0 to GIC-II occurs. Specifically, the value of

fragmentation index in the Altin Mountains is largest with the number over 0.8, indicating the highest degree of

fragmentation. The inferior value of fragmentation index appears in the Karakoram Mountains at 0.41. Both are

observed with a significant increase in glacier number from the RGI 4.0 to GIC-II, whereas the decreases of

glacier area in the period are slight. In addition, the Kunlun Mountains, Qiangtang Plateau, Qilian Mountains,

Hengduan Mountains, Nyainqentanglha Mountains and Tangula Mountains are observed with positive values of

fragmentation index, in which the Qiangtang Plateau and Hengduan Mountains not only have an increasing

glacier number, they also experienced an apparent decrease of area in glaciers from the RGI 4.0 to GIC-II.

Moreover, the Gandise Mountains, Pamir Plateau, and Himalayan Mountains are calculated with negative

fragmentation indexes. The Gandise Mountains went through little decrease in glacier number, while the

decrease of area in glaciers is largest over the studied mountains from the RGI 4.0 to GIC-II.

The ratio of separated glaciers was calculated using Eq. (8) to quantify the glacier number with separation in

each mountain. Approximately 29.7% of glaciers in the Qiangtang Plateau have separated into pieces, which is

the highest ratio of glaciers with separation among all the studied mountains. On average, every glacier in the

Qiangtang from the RGI 4.0 disintegrated into approximately 1.4 sub-glaciers in the GIC-Ⅱ, as calculated using

Eq. (9), which is also the largest number over all mountains. The glaciers formed from the disintegrated glaciers

in the RGI 4.0 account for 10-15% of the total glacier number in GIC-II in more than half of the studied

mountains, while the Pamir Plateau and Himalayan Mountains only contain approximately 3.4% and 4.4% of

split glaciers, respectively. The causes of glacier separation differ from the maritime-type and continental



glaciers. For the maritime glaciers, the ocean current, the strength of wind and self-melting all induce and even
accelerate glacier fracture. In the continental glaciers, topographical, geological and climate changes are the
dominant factors contributing to the deformation of glaciers.

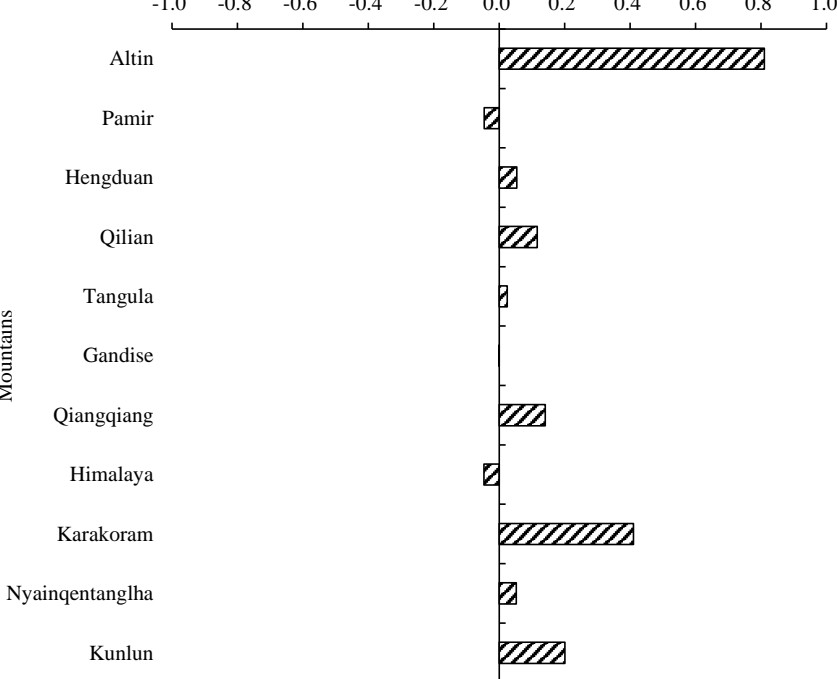


**Fig. 7 Fragmentation indexes of the glaciers in different mountains**

**6 Uncertainties in the recalculated inventories**
**6.1 Uncertainty of input data**
Glaciers in the study area are divided into two types, including maritime and continental glaciers. The greatest
difference in the two types is the summer accumulation- and winter accumulation-dominated patterns,
respectively (Huang, 1990; Shi and Li, 1981). The maritime glaciers are mainly distributed in the southeastern
Tibetan Plateau, while the continental type of glaciers is generally distributed in the other areas. Typically, more
extensive snow and cloud cover exist during ablation seasons (winter for the maritime glaciers and summer for
the continental glaciers), leading to the inconsistency of glacier outline in the same map. This inconsistency is
one source of uncertainty in the recalculated inventories. Regarding the image sources used in this study, some
glaciers (14%) were mapped in winter (November to March), while the remaining 86% of glacier maps were
acquired from April to October (summer). Thus, the technology used to extract the snow and cloud cover from
the original images is important to efficiently determine the ice coverage. The accuracy of glacier delineation is
mainly determined by seasonal snow around the ice margin or within the debris-covered area, and by cloud
cover over the glacier surface. In practice, a value of 2.0 was set as the threshold for TM3/TM5 to differentiate



snow within a five-pixel buffer of the glacier outline and debris-covered area (greater than 2.0), and cloud cover
within the clean-ice area (less than 2.0). Notably, 86% of images have 20% snow/cloud coverage, in which
approximately 48% of images have snow/cloud cover of less than 10%. The lower-quality images (snow/cloud
coverage greater than 20%) are mainly concentrated in the western Himalayan region (30-32°N, 77-81°E) and
Kunlun Mountains (36°N), whereas the inland Tibetan Plateau (33-35°N, 84-90°E) displays the best image
quality (Li, 1986; Pu, 2001; Mi et al., 2002). According to the statistics, 1494 $km^2$ out of the total area of 43087
$km^2$ are debris-covered surfaces (Guo et al., 2015).
On the other hand, many studies have suggested that the vertical accuracy of the TOPO DEM (30 m grid cell)
used in the GIC-I is better than 11 m on glaciers with mean slopes <24° (Chinese National Standard, 2008;
Shangguan et al., 2010; Wei et al., 2015; Xu et al., 2013; Zhang et al., 2016). However, the slopes of more than
$2\times10^4$ $km^2$ of glaciers are greater than 24°, which may result in a larger uncertainty. In addition, the glacier
outlines in the GIC-II were mapped using several different satellite images, including the SRTM DEM acquired
by the radar interferometry with C-band and X-band in early February 2000 (Rabus et al., 2003; Zwally et al.,
2011), the 1 arc-second SRTM C-band DEM, the non-void-filled SRTM C-band DEM with a swath width of 225
km (http://earthexplorer.usgs.gov/) TerraSAR-X (June 2007) and its twin satellite TanDEM-X (June 2010)
launched by the German Aerospace Center (DLR) (Hajnsek et al., 2007), etc. The difference in projection angle,
collection period and pixel layout in data sources are other sources of uncertainty in the recalculated inventories,
which must be improved in the future related studies. In addition, uncertainty is unavoidably bound with using
the extracted slope data from the SRTM DEM 30 m map in 2001 to present the slope distribution in both RGI
4.0 and GIC-Ⅱ (Jiskoot and Mueller, 2012). Further exploration on this uncertainty is also needed.

**6.2 Inconsistency of data source dates in the same glacier map**

Different glacier information was interpreted by different images collected in the 1970s or the 2000s. The direct
generalization and comparison may cause some bias in the results. Specifically, 84.34% of the images in the RGI
4.0 were collected between 1956 and 1980, and 12.27% of them were collected from 1981 to 2008, while 3.37%
of the collected years were missing and one of the images was obtained from 1920. Regarding the data sources
of glaciers in the GIC-II, 84.55% of images were collected in the period from 2004 to 2011. Notably, 15.01% of
images were collected from 1958 to 1980. Additionally, the source dates of the remaining 0.44% of images were
missing. Moreover, the information for a number of glaciers in the southeastern Tibetan Plateau have not been
updated in the GIC-II. However, they were treated as being from the same year in this study to simplify
quantitation. No updated images for a number of glaciers over the southeastern QTP were available in the GIC-
II, and thus the information in the RGI 4.0 was used. Therefore, the comparison of the results between the two
inventories should be interpreted with caution.

**6.3 Inconsistency of the boundary pixel size in glacier volume calculation**

An assumption in volume calculations is that all grids inside a glacier have the same size. In fact, the border of
glaciers is always curved, and thus the boundary grids with the centre inside are partially included while those
without the centre inside are excluded in the glacier thickness estimation. However, these boundary grids are
treated as the same size to obtain the overall average thickness of a glacier. Thus, another source of uncertainty in
the calculation is derived from the boundary pixel size. The error range has been obtained using Eq. (10) to estimate





the impact of uncertainty in glacier area induced by boundary pixels on the calculated glacier volume (Table 5).
The Qiangtang Plateau and Tangula Mountains have the largest errors of over 6% in the RGI 4.0. In the meantime,
both errors are greatest in the GIC-Ⅱ with the values of 6.68% and 5.8%, respectively. However, the Pamir
Plateau has the smallest error in both RGI 4.0 and GIC-Ⅱ at 3.96% and 2.84%, respectively. Most of the errors
in other mountains range from 4~5%.
In addition, another comparison of the equation-based and calculated glacier volumes in the eleven mountains has
been conducted and the results are presented in Table 5. The results of the RGI 4.0 indicate that the majority of
values obtained using empirical equations are consistent with the error ranges of the calculation, in which only the
Qiangtang Plateau displays a slight discrepancy. A small overestimation of the empirical equation exists in the
Qiangtang Plateau. In addition, the glacier volumes in the Altin Mountains, Pamir Plateau, Hengduan, Qilian and
Gandise Mountains are underestimated by the volume-area scaling in the GIC-Ⅱ.

**Table 5 Error estimates of the calculated glacier volumes**

| Mountains | Glacier volume in RGI 4.0 | | | | Glacier volume in GIC-II | | | | |
| --- | --- | --- | --- | --- | --- | --- | --- | --- | --- |
| | Calc_high limit (km³) | Calc_low limit (km³) | Eq. (1) - based values (km³) | Error rate/ % | Calc_ high limit (km³) | Eq. (2) - based high limit (km³) | Calc_ low limit (km³) | Eq. (2) - based low limit (km³) | Error rate/ % |
| Altin | 33.21 | 30.16 | 24.50 | 4.81 | 31.12 | 16.00 | 28.54 | 14.71 | 4.32 |
| Pamir | 259.23 | 239.46 | 234.08 | 3.96 | 289.65 | 171.62 | 273.67 | 161.02 | 2.84 |
| Hengduan | 209.30 | 192.46 | 136.39 | 4.19 | 136.23 | 79.48 | 126.31 | 74.57 | 3.78 |
| Qilian | 194.47 | 176.81 | 140.93 | 4.76 | 141.58 | 87.60 | 130.22 | 81.35 | 4.18 |
| Tangula | 189.91 | 167.53 | 191.64 | 6.26 | 130.76 | 141.96 | 116.43 | 138.54 | 5.80 |
| Gandise | 206.17 | 187.96 | 108.54 | 4.62 | 99.31 | 59.27 | 92.71 | 52.30 | 3.44 |
| Qiangtang | 205.73 | 180.11 | 234.04 | 6.64 | 146.07 | 169.65 | 127.79 | 164.21 | 6.68 |
| Himalaya | 736.09 | 674.49 | 643.63 | 4.37 | 628.31 | 505.74 | 579.52 | 489.57 | 4.04 |
| Karakoram | 528.67 | 474.96 | 554.25 | 5.35 | 547.07 | 623.79 | 502.66 | 554.84 | 4.23 |
| Nyainqent anglha | 1112.14 | 1019.29 | 971.96 | 4.36 | 976.52 | 891.88 | 898.35 | 826.83 | 4.17 |
| Kunlun | 1262.57 | 1152.80 | 1459.86 | 4.54 | 1089.72 | 1174.19 | 998.63 | 1061.73 | 4.36 |


## 7 Challenges and expectations for future studies

Glacier formations in different mountains display substantial discrepancies due to the special topographic,
geological and geomorphologic conditions. These discrepancies result from several crustal movements occurring
from the northern to southern QTP in different historical periods. Therefore, the unified equation is not always
suitable for all of these mountains. Additional field survey results and observations must be collected to improve
the quality of the bedrock elevation database and increase the accuracy of glacier volume calculation. More
accurate surface elevation information is also important. However, a large gap still exists to interpret the





information obtained from the remote sensing images of surging glaciers (Gardner et al., 2013; Kääb et al., 2014;
Neckel et al., 2014). Therefore, the algorithm used to calculate the cloud- and debris-covered areas and derive a
finer glacier outline must be strengthened based on the overwhelming number of remote sensing images and
corresponding products. In addition, due to the lack of field observations of glaciers in the southeastern Tibetan
Plateau, a more complete use of relevant materials to recover the current glacier information and update the GIC-
II are also substantial challenges but important needs. Moreover, uncertainty quantifications must be further
developed. In the future related studies, the error range should also be reduced to more precisely understand the
actual state of the glacier.
**8 Data availability**
The data are available under https://doi.org/10.11888/Glacio.tpdc.270390 (Liu, 2020). For the time of review, the
data will be accessible through the following review link https://data.tpdc.ac.cn/en/data/4b88e394-0eb4-44c4-
aa38-32aeb614daff/.
**9 Conclusion**
We provided a set of recalculated data for all glaciers over the QTP in the RGI 4.0 and GIC-II inventories using a
slope-dependent algorithm based on several elevation datasets. The two recalculated glacier inventories were
compared in the eleven major mountains to investigate glacier changes in the context of climate change during the
past few decades. The main results are summarized below.
(1) The glacier volumes calculated using the slope-dependent algorithm perform better than the traditional
area-volume-based equations. The glacier volumes in the inland Tibetan Plateau have been overestimated by the
traditional method, while the glacier volumes in the western and southern mountains tend to be underestimated.
(2) The value of fragmentation index in the Altin Mountains is largest, indicating the highest degree of
fragmentation. The Karakoram Mountains and Kunlun Mountains have comparably larger fragmentation indexes,
suggesting a stronger effect of climate changes on the glaciers in these mountains.
(3) Most of the surging glaciers are observed in the Karakoram and Kunlun Mountains, while the Gandise
and Himalayan Mountains contain the greatest number of disappeared glaciers during the study period. In addition,
the largest glacier volume loss appears in the Karakoram and Himalayan Mountains. The Karakoram Mountains
also exhibit the largest surged glacier volume.
(4) An obvious offset of glacier volumes between different aspects is observed in most mountains. In general,
the glaciers on the western and southern aspects displayed a greater reduction in volume in the studied period.
Glaciers with increased volumes are mainly located on the northern and northeastern aspects in the northern
mountains, while the southern mountains have surging glacier volumes on the eastern and southeastern aspects.



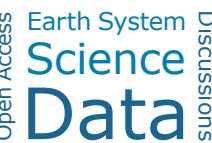

**Author contributions.** Data collection and preprocessing: X.L., H.Y., Z.X., C.G., R., Q.Y.; Supervision: H.Y.,
Z.X.; Writing-original draft: X.L. and H.Y.; Writing-review and editing: X.L., H.Y. and Z.X.
**Competing interests.** The authors declare no conflicts of interest.
**Acknowledgement.** The work is financially supported by the State Key Program of National Natural Science of
China (91647202). The authors are grateful to data providers. We thank Andreas Scheidegger, and Reynold Chow
for the cooperation.

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



**Appendix**

**Table A1 Observed glacier information in the QTP**

| Mountains | Glaciers | LON | LAT | S1 (km$^2$) | S2 (km$^2$) | L1 (m) | L2 (m) | Period | Reference |
|---|---|---|---|---|---|---|---|---|---|
| Qilian | Shule_1 | 97.71 | 38.60 | 13.236 | 12.330 | 7520 | 3305 | 1966-2006 | |
| Qilian | Shule_2 | 97.86 | 38.51 | 10.427 | 9.824 | 6831 | 4135 | 1966-2006 | |
| Qilian | Shule_3 | 98.69 | 38.23 | 15.619 | 16.79 | 6797 | 5505 | 1966-2007 | |
| Qilian | Shule_4 | 97.79 | 38.46 | 15.054 | 14.95 | 7069 | 5932 | 1966-2006 | Gärtner-Roer et al., 2014 |
| Qilian | Shule_5 | 97.31 | 38.70 | 7.749 | 13.88 | 2462 | 4790 | 1966-2006 | |
| Qilian | Shule_6 | 97.24 | 38.71 | 15.414 | 13.82 | 6795 | 6470 | 1966-2006 | |
| Qilian | Shule_7 | 97.22 | 38.75 | 1.576 | 12.18 | 4555 | 2413 | 1966-2006 | |
| Qilian | Bayi | 98.57 | 39.23 | 6.675 | 4.076 | 4830 | 3748 | 1956-2007 | Wang & Pu, 2009 |
| Nyainqentanglha | Zhadang | 90.67 | 30.47 | 1.92 | 1.68 | 2224 | 1451 | 2001-2009 | Zhu et al., 2014 |
| Nyainqentanglha | Gurenhekou | 90.45 | 30.19 | 1.574 | 1.333 | 2834 | 2086 | 2001-2009 | Ma et al., 2008 |

Note: S1, S2, L1, and L2 represent the glacier area in the RGI 4.0 and GIC-II and lengths of glaciers in the RGI 4.0 and GIC-II, respectively.