# Peer review of "Consolidating the Randolph Glacier Inventory and the Glacier 1 Inventory of China over the Qinghai-Tibetan Plateau and 2 Investigating Glacier Changes Since the mid-20th Century 3"

_Earth System Science Data, 2020_

## Referee Comment (RC1) · Anonymous Referee #1 · 1 Oct 2020

This study is presenting a new method to derive glacier volumes and compares several approximations to obtain glacier volumes after application to two different datasets. Resulting volumes are presented at the mountain range level rather than for individual glaciers. Although I think it makes much sense to determine glacier volume changes from two glacier inventories, I have a couple of major issues with this study that I are shortly describe in the following:

(1) The English is not good enough and requires revision by a native speaker. Due to this, it was very stressful for me to read the text and sometimes I have to guess what

the authors could have meant.

(2) As far as I can see, the study introduces a new method of glacier volume calculation (Section 4.1). I think this requires proper introduction (showing results and uncertainties for individual glaciers) in a more topical journal (e.g. The Cryosphere) before it can be applied widely and used for datasets in ESSD. I do also not fully understand how this method is working, as the text describing the method is very short and equations are poorly illustrated (e.g. where in Fig. 2 can I find the variables used in Eqs. (3) and (4) and why is a grid of 1 km used when the SRTM DEM has 30 m resolution?).

(3) Glacier areas in RGI4.0 are highly flawed in this region and are generally too large (e.g. due to missing rock outcrops). They should thus better not be used with methods that are based on an up-scaling of area alone or any change assessment (neither area nor volume). The glacier volume changes calculated here (Table 4) are thus also much too high and basically reflect differences in interpretation rather than real glacier changes.

(4) Also the results for disappeared, fragmented and surged glaciers (Figs. 5 to 7) are strongly impacted by the flaws in the digitization of RGI4.0 and present largely arbitrary results. In my opinion the RGI4.0 dataset is of insufficient quality for such calculations.

(5) It is unclear to me why so many different methods of volume calculation have been applied and which of these are used for which dataset. For example, the authors name it 'Calculated', 'Equation-based' or 'DGA-derived' in Table 2 and Calculated, Equation 1 and Equation 2 in Table 4. Where are they described, which method is used for what purpose?

(6) The authors describe a long list of uncertainties in Section 6, but miss to mention that RGI4.0 has such a bad quality in the study region. I see nowhere in the study a figure showing a glacier outline overlays from both inventories to illustrate the problem.

(7) In effect, it seems the authors present differences between the two inventories
as real changes in glacier number, area and volume and are unaware that these are largely governed by the poor RGI4.0 quality. Its poor geo-location or missing rock outcrops are not even mentioned.

As a remark to L11, I think the QTP is only a part of the 'Third Pole'. The Third Pole also includes regions outside of QTP (e.g. western Pamir/Karakoram and Hindu-Kush). As a short note, I think the scale of the map in Fig. 4 is inappropriate to visualize the differences. Where is the class 'mountains' and where are unfilled boundaries (as in the legend)? Please also note that the two datasets in the Supplement have a different projection and file/attribute names contain characters that cannot be displayed.

---

## Referee Comment (RC2) · Anonymous Referee #2 · 7 Oct 2020

General.

This is an interesting paper that goes into details with a new evaluation of glacier changes in the Quinghai-Tibetan Plateau (QTP). They aim at improving former area and volume estimates of the glaciers and the changes that can be derived from the different glacier inventories. The authors propose a new slope-dependent algorithm to calculate the volumes and show that this approach gives better volume data than the former often used area-volume scaling algorithm. They combine and use input elevation data from different available inventories. This is a useful study. I think they

give improved data about the glacier changes in the region. The QTP region is large and varied both in climate and topography so it is a challenging task to obtain reliable data on glacier area and volume changes. QTP is an important region when comes to water balance studies and changes and this paper is a significant contribution to future analysis of water balance.

The paper is well written. It is fairly easy to follow the language. I am not English native speaker myself, but my impression from the language is good. However, there are some unclear statements and corrections that are needed before publication.

Comments.

Surged glaciers is an important concept in the paper and I think they need to define what is meant by Surged glacier with a paragraph early in the paper, maybe under section 1.3. As it is we suddenly meet surged glaciers for the first time in equation 8, line 335 and below. Surged and disappeared glacier is an important part of the analysis. Definition of surged glaciers is not obvious to the general reader. Surge is a periodic sudden advance of the glacier during a short time period of months to a few years. The glaciers have a long quiescent (up-building period) of several years between each active surge advance. Karakoram, Kunlun and Pamir are regions with high number of surging glaciers.

Line 48. Delete: "led by a distinguished expert in glacier studies in China", It is not appropriate to characterize the authors you refer to. Just write: A study predicted that . . . . . .

Line 331-333. They write: "Meanwhile, the shear stress would also increase and basal sliding would accelerate, which is the key interpretation of how the glacier movement and deformation will develop. ". I do not think this statement is correct, or at least it is more complicated. The basal shear stress depends on both the thickness of the ice and the slope of the glacier surface. When you have more melt and a thinning of the ice the basal shear stress will decrease, however, if the glacier get steeper it will

increase. It is not obvious that the basal sliding will accelerate. Rather opposite in the long run, as the glaciers get thinner, the shear stress will decrease and the basal sliding will decrease. The impact of the dynamics is not a part of this paper anyway so I think they should take out or rewrite these lines. The disintegration of the glaciers which is one of the points of this paper is more related to melting, thinning of the ice and lowering of the glacier surface than to the flow dynamics. Also, in the Abstract, line 32 they write: "Pamir Plateau, which displays the highest trends of glacier movement and deformation. ". I do not understand this statement. Is this based on what they write in line 331-332? If so I think they should rewrite and delete the statement as I said above. See also my comments to lines 522-524 below.

In the paragraph starting at line 372 they discuss GRACE data. They say that GRACE data are chosen to compare and validate the calculated results and products of volume changes as given in Table 2. They say that "An underestimation is observed in the results obtained with the volume-area scaling." But is that compared to GRACE data? This is unclear to me. From Table 2 there are huge differences between equation-based volume change and DGA (Derivations of Gravity Anomaly) volumes. GRACE data is only able to indicate mass changes as average values over quite large areas of about 100X100km and therefore not for individual small glaciers. In the context of this paper it is therefore only useful as a very coarse estimate of mass changes. It can be compared to the average values obtained in the paper to indicate or validate the results, but with very limited or no value down on individual glaciers. It is unclear to understand how the GRACE data is used.

Line 522-524 is unclear. They write: "For the maritime glaciers, the ocean current, the strength of wind and self-melting all induce and even accelerate glacier fracture. In . . ...to the deformation of glaciers." What is self-melting ? And fracture means breaking (like when you get crevasses in the ice) And deformation of glaciers ? I suppose they mean glacier fragmentation or glacier separation. Deformation is related to the flow dynamics. The glacier ice deforms under high pressure, but the deformation will not increase due to climate warming and shrinking glaciers, rather decrease as the glaciers get thinner. Maybe they could write: "For the maritime glaciers, the changes in ocean currents (affecting the precipitation pattern), the strength of wind and increased surface melting of the glaciers all induce and even accelerate glacier thinning and thus disintegration. In the continental glaciers, topographical, geological and climate changes are the dominant factors contributing to the disintegration of glaciers."

However, changes in ocean currents, wind changes and surface melt are all effects of climate changes so both maritime and continental glaciers are affected by climate changes, but the continental dominated by air temperature changes.

They use in general very precise numbers, as in line 401 to 403. It looks strange to me to write: "...is approximately 54874.79 km2" as in line 401. This is a very precise number, even given with two decimals, thus it is not "approximately". There are many similar examples of very precise numbers in the paper. There are large uncertainties in the RGI 4.0 so it does not make sense to give such exact numbers.

The reference list is fine. However, in line 74 they refer to Machereet et al. (1988). This reference is not in the reference list.

Figures and Tables are in mostly clear and useful.

Figure 1 shows the regions and elevation pattern, but why do they use so precise numbers as elevation from 84 m to 8299 m ? Why not just use 100 m to 8300m. In the captions they give length and width for some regions, but area, length and width for others; why not area, length and width for all?

Figure 4 shows mountain regions with surged and disappeared glacier. However, this figure is impossible to read. Even when I enlarge the figure in the pdf-file to 400% it is hard to get any readable information out of it. I would suggest to take out that figure. Or maybe replace it by a close up of one region with both surging and disappeared glaciers.

---

## Author Comment (AC1) · 1 Nov 2020

Dear the Editor and Reviewer Thank you very much for giving us the opportunity to revise and improve our manuscript. Many thanks also to your valuable comments. We have revised our manuscript accordingly. The revised text is in red in the manuscript. A point-to-point response to all the comments is provided below. The comments are copied in black text. Our responses are in red text.

Responses to Reviewer #1 Evaluations:

1. The English is not good enough and requires revision by a native speaker. Due to this, it was very stressful for me to read the text, and sometimes, I have to guess what the authors could have meant. Response: We are terribly sorry that we didn't find a suitable native speaker for helping check English writing this time. But we are still trying to find such help in near future.

2. As far as I can see, the study introduces a new method of glacier volume calculation (Section 4.1). I think this requires proper introduction (showing results and uncertainties for individual glaciers) in a more topical journal (e.g. The Cryosphere) before it can be applied widely and used for datasets in ESSD. I do also not fully understand how this method is working, as the text describing the method is very short and equations are poorly illustrated (e.g. where in Fig.2 can I find the variables used in Eqs. (3) and (4) and why is a grid of 1 km used where the SRTM DEM has 30 m resolution?). Response: Thanks for your valuable suggestion. First, we're sorry for the poor correspondence between Fig.2 and Eqs. (3) and (4). In fact, the number in the subscript of variables corresponds to the grid number in Fig.2. Specifically, $Z_0$, $slope_0$, $x_0$, $y_0$ are the elevation, slope, longitude and latitude, respectively, in the grid with number 0. $Z_{1,i}$, $slope_{1,i}$, $x_{1,i}$, $y_{1,i}$ are the elevation, slope, longitude and latitude, respectively, in the grid with number 1. $Z_{2,j}$, $slope_{2,j}$, $x_{2,j}$, $y_{2,j}$ are the elevation, slope, longitude and latitude, respectively, in the grid with number 2. i represents how many the neighboring grids of the grid with number 0 are. j is how many the neighboring grids of the grid with number 1 are. To make it clear, relevant description has been added in Lines 322-326 in the revised manuscript. In this study, the elevation data in the grid of 625 m×625 m was used and resampled from the SRTM DEM with 30 m resolution by using "nearest" technique. The relevant description was shown in Lines 227-230 in the original manuscript.

3. Glacier areas in RGI 4.0 are highly flawed in this region and are generally too large (e.g. due to missing rock outcrops). They should thus better not to be used with methods that are based on an up-scaling of area alone or any change assessment (neither

area nor volume). The glacier volume changes calculated here (Table 4) are thus also much too high and basically reflect differences in interpretation rather than real glacier changes. Response: Thanks for your valuable comment. The flawed glacier areas (including missing rock outcrops) in RGI 4.0 don't affect the results of glacier volume with the proposed algorithm in this study, which means the calculated volume is equal to the actual volume in the glacier with missing rock outcrops. The specific explanation can be shown by the equations as follows.

where , and A, A0 mean the thickness and area, respectively, of the studied glacier with and without grids at missing rock outcrops. N, N0 are the number of all grids and the number of grids at missing rock outcrops, respectively, within the studied glacier. a is the area of one grid, which is approximately equal to 0.39 km2 (625 m×625 m) in this study.

4. Also, the results for disappeared, fragmented and surged glaciers (Figs. 5 to 7) are strongly impacted by the flaws in the digitization of RGI 4.0 and present largely arbitrary results. In my opinion the RGI 4.0 dataset is of insufficient quality for such calculations. Response: Yes, the flaw in the digitization of RGI 4.0 is a main source of uncertainty in disappeared, fragmented and surged glaciers. However, the RGI 4.0 provides valuable baseline information of glaciers for evolution exploration, so the dataset is prior to be considered in such researches. Another investigation manifests that in the compilation of RGI 4.0, the GIC-âĚă has been used to manually match the glaciers in Chinese territory to validate the locations, approximately 80% of them being in the QTP. In the process, a separation not exceeding 2 km was found sufficiently to match the glacier in GIC-âĚă with its closest RGI 4.0 counterpart. The results show that 38% of them are exactly matched, 43% have separations within 300 m between the RGI 4.0 and GIC-âĚă. Only 1.4% of glaciers fail the 2-km test of proximity. The relevant description has been added in '6.1 Uncertainty of input data' in the revised manuscript (Lines 586-595).

5. It is unclear to me why so many different methods of volume calculation have been applied and which of these are used for which dataset. For example, the authors name

it 'Calculated', 'Equation-based' or 'DGA-derived' in Table 2 and Calculated, Equation 1 and Equation 2 in Table 4. Where are they described, which method is used for what purpose? Response: The empirical Equation 1 and Equation 2 recommended by the compilers in GIC-âĚă and GIC-âĚą, respectively, were obtained by area-volume scaling based on observations in 1970s and 2000s. Both results are "Equation-based". "Calculated" values are computed by the proposed method in this study. The DGA (the derivations of gravity anomaly) data (Liu et al., 2016) are a sum of changes in soil moisture and glacier volume over the QTP from 2003 to 2010 on the grids with spatial resolution of 1°, which were sourced from Gravity Recovery and Climate Experiment outputs (GRACE) (Liu et al., 2015, 2016). Soil moisture data with spatial resolution of 0.25° were extracted from the Global Land Data Assimilation System (GLDAS) products (Hiroko and Rodell, 2016) during the same period to obtain the changes in glacier volumes included in the DGA dataset (DGA-derived results). In detail, the DGA-derived results are the changes in glacier volume in $1°\times1°$ pixels calculated by subtracting the DGA value from the corresponding GLDAS soil moisture value (resampled from the $0.25°\times0.25°$ to the $1°\times1°$ pixel), and used to compare with the recalculated and traditional equation-based glacier volumes by integrating the individual glacier volume into corresponding $1°\times1°$ pixel.

6. The authors describe a long list of uncertainties in Section 6, but miss to mention that RGI 4.0 has such a bad quality in the study region. I see nowhere in the study a figure showing a glacier outline overlays from both inventories to illustrate the problem. Response: We have added two aspects of the uncertainty in the RGI 4.0. On the one hand, the uncertainty from the digitization of RGI 4.0 has been mentioned in the response to Comment 4. On the other hand, a big part of glacier information was sourced from the Central Asia (Region 13) containing 25 nominal glaciers (24 km2). It is another uncertainty source. Relevant description has been added in Lines 593-595 in the revised manuscript. A schematic map for uncertainty estimation has been drawn as Fig. 3 in the revised manuscript., The description as "Provided the grids of center within a glacier are calculated, the minimum and maximum of real glacier

area are designed as deducting and including the half area of boundary grids from the grid-based sum of area." has been correspondingly added in Lines 352-354.

Fig. 3 Schematic map for uncertainty estimation Note: The area surrounded by the blue curve is a sketched glacier. The grids with "âŮŃ" and thicker black border line are boundary pixels. The area shaded by left-inclined black lines and right-inclined red lines present unreal-calculated and real-uncalculated part of area, respectively.

7. In effect, it seems the authors present differences between the two inventories as real changes in glacier number, area and volume and are unaware that these are largely governed by the poor RGI 4.0 quality. Its poor geo-location or missing rock outcrops are not even mentioned. Response: Thanks for this valuable comment. As for the uncertainty from poor geo-location in the RGI 4.0, the results of manual checking between the RGI 4.0 and GIC-âĚă have been given in the response to Comment 4. Regarding missing rock outcrops, we haven't found such information in the technical report of the RGI 4.0. However, even a number of missing rock outcrops included in the RGI 4.0, the result of glacier volume calculated by the proposed method in this study is not affected as the response to Comment 5.

8. As a remark to L11, I think the QTP is only a part of the 'Third Pole'. The Third Pole also includes regions outside of QTP (e.g. western Pamir/Karakoram and Hindu-Kush). As a short note, I think the scale of the map in Fig. 4 is inappropriate to visualize the differences. Where is the class 'mountains' and where are unfilled boundaries (as in the legend)? Please also note that the two datasets in the Supplement have a different projection and file/attribute names contain characters that cannot be displayed. Response: The difference in projected coordinate system between the two datasets was considered in the manuscript. In this process, the geographic coordinate system GCS_WGS_1984 was employed to display the two datasets in the same map. The class 'mountains' in Fig.4 has been explained in Note and added as "Note: 1-Altin Mountains; 2-Pamir Plateau; 3-Hengduan Mountains; 4-Qilian Mountains; 5-Tangula Mountains; 6-Gandise Mountains; 7-Qiangtang Plateau; 8-Himalayan Mountains; 9-

Karakoram Mountains; 10-Nyainqentanglha Mountains; and 11-Kunlun Mountains." in Lines 469-471 in the revised manuscript. In addition, we didn't find unfilled boundaries as you mentioned. Either, we didn't find the file/attribute names containing characters that cannot be displayed from our side. Could you please specify where the problems are, so then we can try to update them at another time?

Please also note the supplement to this comment:
https://essd.copernicus.org/preprints/essd-2020-71/essd-2020-71-AC1-supplement.pdf

[Figure]

**Fig. 1.** Schematic map for uncertainty estimation

---

## Author Comment (AC2) · 1 Nov 2020

Dear the Editor and Reviewer Thank you very much for giving us the opportunity to revise and improve our manuscript. Many thanks also to your valuable comments. We have revised our manuscript accordingly. The revised text is in red in the manuscript. A point-to-point response to all the comments is provided below. The comments are copied in black text. Our responses are in red text.

Responses to Reviewer #2 Evaluations:

[Figure]

1. Surged glacier is an important concept in the paper and I think they need to define what is meant by Surged glacier with a paragraph early in the paper, maybe under section 1.3. As it is we suddenly meet surged glaciers for the first time in equation 8, line 335 and below. Surged and disappeared glacier is an important part of the analysis. Definition of surged glaciers is not obvious to the general reader. Surge is a periodic sudden advance of the glacier during a short time period of months to a few years. The glaciers have a long quiescent (up-building period) of several years between each active surge advance. Karakoram, Kunlun and Pamir are regions with high number of surging glaciers. Response: Thanks for your valuable suggestion. The definition of surged glaciers has been added as "If a glacier occurs advancing from RGI 4.0 to GIC-âĚą, it is defined as a surged glacier. In detail, surge is a periodic sudden advance of the glacier during a short time period of months to a few years. The glaciers have a long quiescent (up-building period) of several years between each active surge advance" in Lines 125-128, and basic information related to the mountains has been added as "Karakoram, Kunlun and Pamir are regions with high number of surging glaciers." in Lines 140-141 in the revised manuscript.

2. Delete: "led by a distinguished expert in glacier studies in China", It is not appropriate to characterize the authors refer to. Just write: A study predicted that ... Response: Thanks for your correction. The sentence has been changed accordingly in Line 48 in the revised manuscript.

3. Line 331-333. They write: "Meanwhile, the shear stress would also increase and basal sliding would accelerate, which is the key interpretation of how the glacier move-ment and deformation will develop.". I do not think this statement is correct, or at least it is more complicated. The basal shear stress depends on both the thickness of the ice and the slope of the glacier surface. When you have more melt and a thinning of the ice the basal shear stress will decrease, however, if the glacier gets steeper it will increase. It is not obvious that the basal sliding will accelerate. Rather opposite in the long run, as the glaciers get thinner, the shear stress will decrease and the basal

sliding will decrease. The impact of the dynamics is not a part of this paper anyway so I think they should take out or rewrite these lines. The disintegration of the glaciers which is one of the points of this paper is more related to melting, thinning of the ice and lowering of the glacier surface than to the flow dynamics. Also, in the Abstract, line 32 they write: "Pamir Plateau, which displays the highest trends of glacier movement and deformation.". I do not understand this statement. Is this based on what they write in line 331-332? If so I think they should rewrite and delete the statement as I said above. See also my comments to lines 522-524 below. Response: Thanks for your correction. The sentence in the Abstract has been revised as "Pamir Plateau, which displays the highest trends of glacier disintegration." in Line 32. The statement in Lines 331-332 in the original manuscript as you mentioned was removed, and the sentence as "Thus, a higher fragmentation index explains a larger possibility in the disintegration of glaciers." has been added in Lines 339-340 in the revised manuscript.

4. In the paragraph starting at line 372 they discuss GRACE data. They say that GRACE data are chosen to compare and validate the calculated results and products of volume changes as given in Table 2. They say that "An underestimation is observed in the results obtained with the volume-area scaling." But is that compared to GRACE data? This is unclear to me. From Table 2 there are huge differences between equation-based volume change and DGA (Derivations of Gravity Anomaly) volumes. GRACE data is only able to indicate mass changes as average values over quite large areas of about 100×100 km and therefore not for individual small glaciers. In the context of this paper it is therefore only useful as a very coarse estimate of mass changes. It can be compared to the average values obtained in the paper to indicate or validate the results, but with very limited or no value down on individual glaciers. It is unclear to understand how the GRACE data is used. Response: We are sorry for this confusion. The Gravity Recovery and Climate Experiment outputs (GRACE) (Liu et al., 2015, 2016) were derived to produce the derivations of gravity anomaly (DGA) data by Liu et al. (2016). In detail, the DGA data are a sum of changes in soil moisture and glacier volume over the QTP from 2003 to 2010 on the grids with spatial resolution of 1°. To

extract the glacier volume from DGA data, soil moisture data with spatial resolution of $0.25°$ were extracted from the Global Land Data Assimilation System (GLDAS) products (Hiroko and Rodell, 2016). Then the glacier volume was obtained by subtracting soil moisture (resampled from the $0.25°\times0.25°$ to the $1°\times1°$ pixel) from DGA data and called DGA-derived results in this study. The DGA-derived data have a resolution of $1°$, and were used to compare with the recalculated and equation-based results integrated by individual glacier volume within corresponding $1°\times1°$ pixels.

5. Line 522-524 is unclear. They write: "For the maritime glaciers, the ocean current, the strength of wind and self-melting all induce and even accelerate glacier fracture. In . . . to the deformation of glaciers". What is self-melting? And fracture means breaking (like when you get crevasses in the ice) And deformation of glaciers? I suppose they mean glacier fragmentation or glacier separation. Deformation is related to the flow dynamics. The glacier ice deforms under high pressure, but the deformation will not increase due to climate warming and shrinking glaciers, rather decrease as the glaciers get thinner. Maybe they could write: "For the maritime glaciers, the changes in ocean currents (affecting the precipitation pattern), the strength of wind and increased surface melting of the glaciers all induce and even accelerate glacier thinning and thus disintegration. In the continental glaciers, topographical, geological and climate changes are the dominant factors contributing to the disintegration of glaciers." However, changes in ocean currents, wind changes and surface melt are all effects of climate changes so both maritime and continental glaciers are affected by climate changes, but the continental dominated by air temperature changes. Response: Thanks for your correction. The sentence has been changed to "For the maritime glaciers, the ocean current (affecting the precipitation pattern), the strength of wind and increased surface melting of the glaciers all induce and even accelerate glacier thinning and thus disintegration. In the continental glaciers, topographical, geological and air temperature changes are the dominant factors contributing to the disintegration of glaciers." in Lines 545-548 in the revised manuscript.

6. They use in general very precise numbers, as in line 401 to 403. It looks strange to me to write: "... is approximately 54874.79 km2" as in line 401. This is a very precise number, even given with two decimals, thus it is not "approximately". There are many similar examples of very precise numbers in the paper. There are large uncertainties in the RGI 4.0 so it does not make sense to give such exact numbers. Response: Thanks for your valuable comment. As for Lines 401-403 in the original manuscript, the results are initial values of statistics. To include the uncertainty in the results, the error estimates have been included. The relevant descriptions are added in the "Abstract" and "6 Uncertainties in the recalculated inventories". Specifically, the description has been revised as "The comparison of the two inventories reveals a total area of glaciers in the QTP of 54874.79±2207.23 km2 in the RGI 4.0 and 43745.48±1707.62 km2 in the GIC-âĚą. The total glacier volume is 4045.81±170.76 km3 in the GIC-âĚą compared with 4716.76±220.72 km3 in the RGI 4.0." in Lines 26-28. The sentences have been added as "Considering the uncertainty from the inconsistency in size of boundary pixels, the error estimates of calculated glacier volume in Table 5 and the error of glacier area estimated by Eq. (10) are included. The results indicate a total area of glaciers of 54874.79±2207.23 km2, 43745.48±1707.62 km2 in the QTP, respectively, in the RGI 4.0 and GIC-âĚą. The total glacier volume changes from 4716.76±220.72 km3 in the RGI 4.0 to 4045.81±170.76 km3 in the GIC-âĚą." in Lines 625-629 in the revised manuscript.

7. The reference list is fine. However, in line 74 they refer to Machereet et al. (1988). This reference is not in the reference list. Response: Thanks for your careful correction. I'm sorry for the writing error. The citation should be "Macheret et al. (1988)" and has been modified in Line 74. The specific reference is shown as follows and has also been added in Lines 830-832 in the revised manuscript. Macheret, Y. Y., Cherkasov, P. A., Bobrova, L. I.: Tolschina i ob'em lednikov djungarskogo alatau po danniy aeroradiozondirovaniya, Materialy Glyatsiologicheskikh Issledovanii: Khronika, Obsuzhdeniya, 62, 59-71, 1988. [in Russian]

8. Figure 1 shows the regions and elevation pattern, but why do they use so precise numbers as elevation from 84 m to 8299 m? Why not just use 100 m to 8300 m. In the captions they give length and width for some regions, but area, length and width for others; why not area, length and width for all? Response: The legend has been changed as you suggested in Fig.1. The caption of Fig.1 has also been added as follows in the revised manuscript. Note: 1-Altin Mountains (area: $6.23 \times 104$ km2; length: 730 km; width: 100 km); 2-Pamir Plateau (area: $2.45 \times 105$ km2; length: 260 km; width: 50-100 km); 3-Hengduan Mountains (area: $3.42 \times 105$ km2; length: 900 km); 4-Qilian Mountains (area: $1.74 \times 105$ km2; length: 800 km; width: 200-400 km); 5-Tangula Mountains (area: $1.72 \times 105$ km2; length: 700 km; width: 150 km); 6-Gandise Mountains (area: $1.49 \times 105$ km2; length: 1100 km; width: 60-100 km); 7-Qiangtang Plateau (area: $4.46 \times 105$ km2; length: 1200 km; width: 760 km); 8-Himalayan Mountains (area: $2.16 \times 105$ km2; length: 2450 km; width: 200-350 km); 9-Karakoram Mountains (area: $9.45 \times 104$ km2; length: 800 km; width: 240 km); 10-Nyainqentanglha Mountains (area: $1.73 \times 105$ km2; length: 1400 km; width: 80 km); and 11-Kunlun Mountains (area: $7.3 \times 105$ km2; length: 2500 km; width: 130-200 km) (Guo, 2011).

9. Figure 4 shows mountain regions with surged and disappeared glacier. However, this figure is impossible to read. Even when I enlarge the figure in the pdf-file to 400% it is hard to get any readable information out of it. I would suggest to take out that figure. Or maybe replace it by a close-up of one region with both surging and disappeared glaciers. Response: Thanks for your valuable suggestion. To generally show the distribution of surged and disappeared glacier over the study area, the Fig. 4 in the original manuscript remains. In the revised manuscript, we have extracted two sub-regions from Karakoram and Gandise Mountains to show the amplification of surged and disappeared glaciers, respectively. The specification of the revised Fig. 4 is shown as follows.

S1 S2 Fig. 4 Disappeared and surged glaciers from the 1970s to 2000s over the QTP Note: 1-Altin Mountains; 2-Pamir Plateau; 3-Hengduan Mountains; 4-Qilian Mountains; 5-Tangula Mountains; 6-Gandise Mountains; 7-Qiangtang Plateau; 8-Himalayan Mountains; 9-Karakoram Mountains; 10-Nyainqentanglha Mountains; and 11-Kunlun Mountains. In addition, S1 is extracted from the Karakoram Mountains, in which glacier advancing typically occurs. S2 is taken out from the Gandise Mountains having the largest loss of glacier volume. In S1 and S2, the blue polygons, red polygons represent disappeared and surged glaciers, respectively. The cyan polygons with a black border are the unchanged part of glaciers between the two datasets.

Please also note the supplement to this comment:
https://essd.copernicus.org/preprints/essd-2020-71/essd-2020-71-AC2-supplement.pdf

**elevation/m**

8300

100

11

10

0    340    680 km

**Fig. 1.** Location and surface elevation pattern of the QTP in China

[Figure]

Fig. 4 Disappeared and surged glaciers from the 1970s to 2000s over the QTP

**Fig. 2.** Disappeared and surged glaciers from the 1970s to 2000s over the QTP